# Modeling human HSV infection via a vascularized immune-competent skin-on-chip platform

Sijie Sun[1,2], Lei Jin[2], Ying Zheng [3,4] & Jia Zhu [1,2,4] ✉

Herpes simplex virus (HSV) naturally infects skin and mucosal surfaces, causing lifelong recurrent disease worldwide, with no cure or vaccine. Biomimetic human tissue and organ platforms provide attractive alternatives over animal models to recapitulate human diseases. Combining prevascularization and microfluidic approaches, we present a vascularized, three-dimensional skin-on-chip that mimics human skin architecture and is competent to immune-cell and drug perfusion. The endothelialized microvasculature embedded in a fibroblast-containing dermis responds to biological stimulation, while the cornified epidermis functions as a protective barrier. HSV infection of the skin-on-chip displays tissue-level key morphological and pathophysiological features typical of genital herpes infection in humans, including the production of proinflammatory cytokine IL-8, which triggers rapid neutrophil trans-endothelial extravasation and directional migration. Importantly, perfusion with the antiviral drug acyclovir inhibits HSV infection in a dose-dependent and time-sensitive manner. Thus, our vascularized skin-on-chip represents a promising platform for human HSV disease modeling and preclinical therapeutic evaluation.

Herpes simplex virus type 1 and 2 (HSV-1 and HSV-2) cause life-long oral and genital recurrent disease and affect >3.7 billion people worldwide[1,2]. Both HSV-1 and HSV-2 can cause neonatal herpes, resulting in high mortality and long-term neurological impairment, while genital HSV-2 infection is also a leading risk factor for increased HIV acquisition and transmission[3–5]. Standard acyclovir antiviral suppressive therapy that limits HSV disease severity and shortens outbreak duration does not reduce HIV risk and faces increasing drug resistance[6–8]. Candidate vaccines that show promise in murine models have failed in human clinical trials[9], thus demanding alternative platforms for understanding HSV pathogenesis in the human system.

Deciphering in vivo multifaceted and collaborative host responses to HSV infection in humans remains challenging, despite our great efforts and the invaluable insights gained from sequential skin biopsies obtained from HSV-2 infected individuals[10–14]. Access to HSV-2-affected human skin requires skillful clinicians and dedicated medical spaces, and is clinically restrained in size, number, location, and timing. Further, the self-reported nature of an outbreak makes it difficult to capture early events of virologic and immunologic importance (<48 h), which are critical in determining disease outcomes. As such, the early kinetics of the immune response and mechanisms of defense are poorly defined in human HSV infection, contributing to knowledge gaps in human immune protection against viral infection.

Bio-inspired and micro-engineered 3-D human tissue and organ models provide attractive alternatives for bridging the gap between human and animal models[15–18]. Recent advancements in tissue engineering, biomaterials, and microfluidics have made it possible to generate in vitro platforms to recapitulate complex tissue architecture with a functional interface and specific microenvironment. Skin, the largest organ of the human body, is comprised of epidermis, dermis,

[1]Department of Laboratory Medicine and Pathology, University of Washington School of Medicine, Seattle, USA. [2]Vaccine and Infectious Disease Division, Fred Hutchinson Cancer Center, Seattle, USA. [3]Department of Bioengineering, University of Washington, Seattle, USA. [4]Institute of Stem Cell and Regenerative Medicine, University of Washington, Seattle, USA. ✉e-mail: jiazhu@uw.edu

and subcutaneous fat to form a three-layer physical barrier that protects against pathogens and harmful environmental substances[19]. The epidermis consists of several stratified layers, from the outermost stratum corneum, stratum granulosum, stratum spinosum, to the innermost stratum basale, while the dermis contains collagen and elastin for mechanical support and elasticity and hosts a network of cells for physiological functions and immune defense. The development of a full-thickness human skin-equivalent using fibroblasts and keratinocytes was first reported by Bell et al. in the early 1980s[20]. Since then, several in vitro skin models have been constructed and commercialized that mimic both dermal and epidermal compartments[21,22]. Recent advances have focused on incorporating immune cells, melanocytes, endothelial cells, adipocytes, dermal papilla cells, and/or sensory neurons to imitate the complexity of real human skin in its composition, structure, and function[23–28].

Skin contains a rich network of blood vessels, and vascularization has been a critical hurdle in bioengineering physiological skin constructs for accurate disease modeling and drug evaluation[29]. Early efforts to incorporate vasculature were achieved by co-culture of endothelial cells in the dermal compartment with growth factors[30,31]. The resulting spontaneously formed capillary-like structure, although it was not perfusable and was randomly distributed, allowed for proper and long-term structure and function of the 3D skin constructs. Recent studies introduced pre-organized vascular scaffolds through decellularization, micropatterning, and 3-D printing to generate vasculature in 3-D skin models[32–38]. Nevertheless, properly functional vascularization that achieves relevant endothelial barrier and subcutaneous permeability remains a major challenge, limiting the development of a realistic system suitable for modeling dynamic host interactions in skin-related diseases and evaluating drug efficacy[39,40]. Here, we report the establishment of a gravity-driven, microfluidic-based, full-thickness human skin-on-chip platform that incorporates an endothelialized, perfusable microvascular network for modeling HSV infection, host immune response, and antiviral drug efficacy.

## Results

### Bioengineering a vascularized biomimetic 3-D skin-on-chip microfluidic device

We utilized skin-specific primary human cells, coupled with lithography-based vascular engineering technology, to fabricate a skin-mimetic microfluidic device composed of a stratified epidermis, an underlying dermis with a collagen-rich extracellular matrix containing fibroblasts and a microvascular network (Fig. 1a, b and Supplementary Figs. 1 and 2). Primary human dermal fibroblasts were first embedded into a native type I collagen gel (7.5 mg/ml) to form a dermal compartment and provide mechanical stability in a plexiglass cassette (Supplementary Fig. 1). A microchannel network (100 μm in diameter) was fabricated using soft lithography and injection molding techniques[41] within the fibroblast-containing collagen and sealed with a collagen-coated glass coverslip. Primary human dermal microvascular endothelial cells were seeded into the microchannel networks through perfusion, to allow for cell attachment on the channel surface and formation of a microvascular bed (Supplementary Fig. 2). After 2 days of culture, a robust endothelium formed, lining the luminal wall. Primary human epidermal keratinocytes were then seeded on top of the fibroblast-containing collagen matrix. When keratinocytes reached confluency (~2 days), they were exposed to an air–liquid interface for an additional 7–10 days to further differentiate into a multilayered stratum epidermis. The 3-D reconstituted confocal image depicts the epithelial layer and vascularized dermis (Fig. 1c).

### Characterization of the 3-D vascularized skin-on-chip

We examined key epithelial marker expression and performed hematoxylin and eosin (H&E) staining for morphological analysis to evaluate the composition of the epithelium formed in the skin-on-chip. Keratin

14 (K14), a basal keratinocyte marker, showed polarized expression exclusively at the basal layer of the epithelium, indicating stratum basale formation (Fig. 1d). The upper layers of keratinocytes lost K14 expression and displayed reduced cellular density. The outermost segment was made of layers of flattened cells that lacked nuclei, suggesting the formation of anucleate cornified epithelium (Fig. 1d, e). Further characterization showed the expression patterns of filaggrin, loricrin, involucrin, and Ki67 in the skin-on-chip were similar to native human skin, indicating keratinocyte terminal differentiation and the formation of stratum cornium, stratum granulosom, stratum spinosum, and stratum basale, respectively (Supplementary Fig. 3). Therefore, the engineered 3-D skin-on-chip has successfully recapitulated a stratified, fully differentiated epidermis.

The dermal compartment in the 3-D skin-on-chip was made-up of fibroblasts, collagen, and an endothelialized vasculature. We found that fibroblasts spread evenly in the perivascular space, as indicated by Phalloidin staining of filamentous actin (F-actin) (Fig. 1f). They also interacted directly with the endothelium, forming a pericyte-like physical association with blood vessels (Fig. 1g). Importantly, confocal imaging revealed an oval-shaped cross-section of the microvasculature (Fig. 1g), despite a square shape in the design and construction. The newly formed oval, tube-like lumen suggests active interactions between fibroblasts and endothelial cells in reshaping the surrounding collagen matrix. We also observed the formation of robust endothelial cell junctions at cell–cell contact regions, indicated by high expression of CD31 and VE-cadherin in the newly formed vessel (Fig. 1g and Supplementary Fig. 4). To verify the integrity and barrier function of the vasculature, we perfused the network with FITC-Dextran (40 kilodaltons [kDa]) (Fig. 1h and Supplementary Movies 1 and 2). The acellular channels showed rapid diffusion of FITC-Dextran throughout the collagen matrix, while endothelialized vessel network displayed resistance (Supplementary Fig. 5), with an apparent permeability of $(1.83 \pm 0.41) \times 10^{-6}$ cm/s, which is similar to isolated mammalian venules $(2 \times 10^{-6}$ cm/s)[41–44]. In addition, the vessel-lining endothelial cells identified by F-actin staining showed cytoskeleton alignment consistent with the flow direction during perfusion, suggesting the newly formed endothelium was responsive to the mechanical flow shear force (Fig. 1i). When perfused with medium containing vascular endothelial growth factor (VEGF) and fibroblast growth factor (FGF), endothelial cells grew tubular capillaries and sprouted from the vascular network upon angiogenic stimulation (Fig. 1j). Overall, these engineered microvessels in the skin-on-chip demonstrate integrity, competence, and functionality in response to mechanical and biological stimulation. Taken together, our microfluidic 3-D skin-on-chip recapitulates key structures of human skin, with a stratum epidermis composed of differentiated keratinocytes and a functional, venule-like microvascular network capable of perfusion.

### HSV infection of the 3-D vascularized skin-on-chip

To further characterize keratinocyte stratification and their susceptibility to HSV infection, we monitored, from the time of keratinocyte seeding to a fully differentiated multilayer, the epidermis histomorphology, biomarker expression, and viral infectivity by directly applying infectious viral particles onto the epidermal surface of the skin-on-chip (Fig. 2a). H&E staining and marker expression revealed that keratinocytes differentiated into keratin 10 (K10) positive stratum spinosum as early as day 3 post air lifting to air–liquid interface, and further developed into a cornified epidermis by day 7 (Fig. 2b, c). By day 9, a fully stratified epidermis expressed markers of keratinocytes early and later differentiation (K14 and K10, respectively) and marker indicative of basement membrane formation (Col IV), comparable to a native epidermis (Fig. 2d). Dermal fibroblasts expressed similar levels and pattern of vimentin (VIM) in bioengineered and native human skin (Fig. 2d). Neither HSV-1 strain K26, a recombinant virus expressing GFP protein, nor HSV-2 strain 186, a virulent wild type strain, yielded active viral

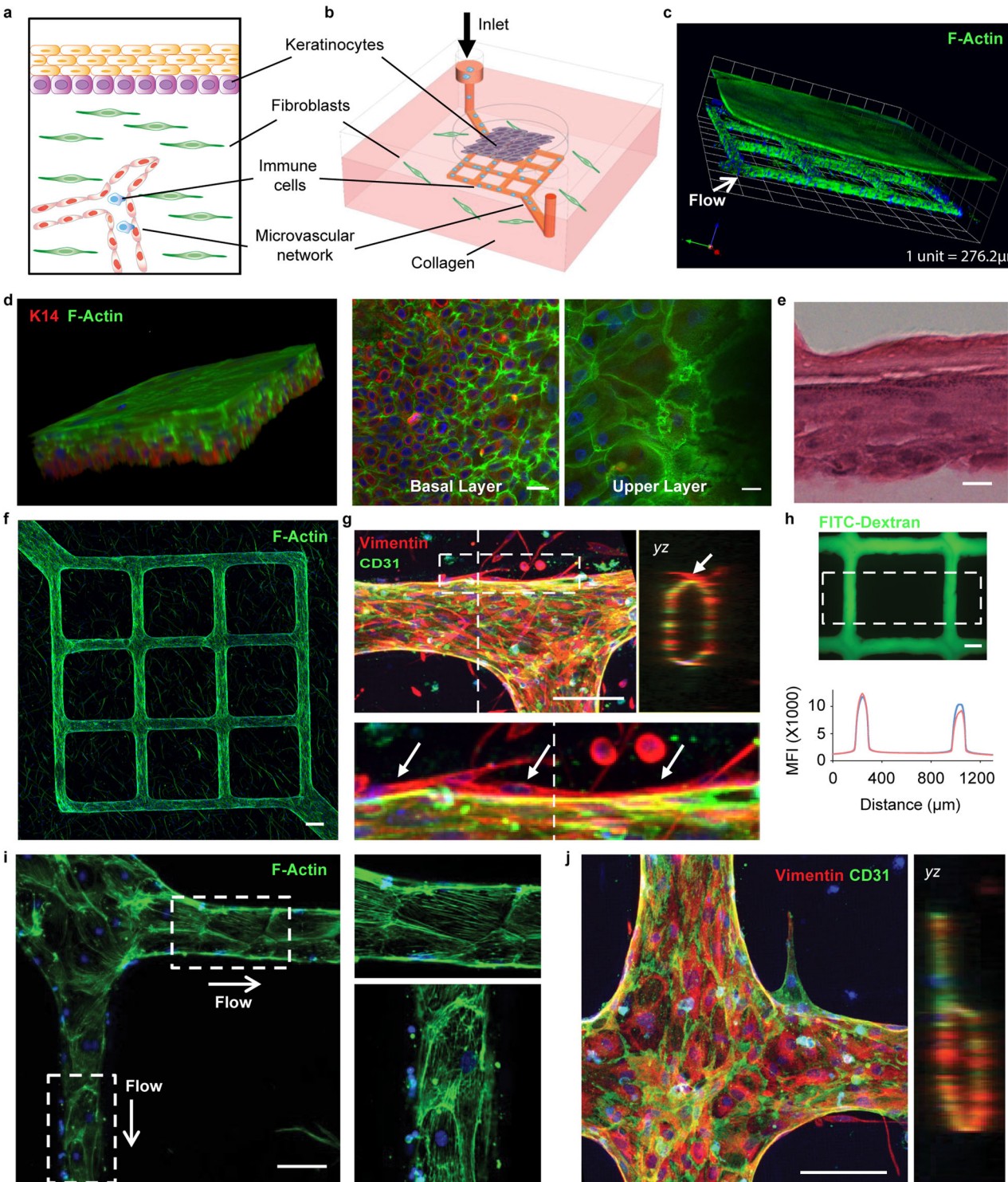

**Fig. 1 | Skin-on-chip contains a stratified epidermis and dermis with a functional microvascular network. a**, **b** Schematics of the major components in native human skin and skin-on-chip. **c** Representative 3-D reconstructed confocal image showing an overview of the cytoskeleton of the micro-engineered epidermis and dermis with underlying endothelialized microvascular network (white arrow indicates microfluidic flow direction). F-actin for cell cytoskeleton (green), DAPI for cell nucleus (blue). **d** Representative 3-D reconstructed confocal image illustrating basale (bottom, K14⁺) and corneous (upper, anucleate) layers in the epidermis of skin-on-chip. F-actin (green), K14 (red), DAPI (blue). Scale bar: 20 μm. **e** A cross-sectional view of hematoxylin−eosin (H&E) stained epidermis in the skin-on-chip. Scale bar: 20 μm. **f** Maximum intensity projected confocal image of an endothelialized microvascular network in the dermis after two weeks of culturing. F-actin (green), DAPI (blue). Scale bar: 100 μm. **g** Projected confocal images showing endothelium marker (CD31) expression, lumen formation (*yz*), and fibroblast capping of endothelial vessel (bottom). CD31 (green), Vimentin (red), DAPI (blue). Scale bar: 100 μm. **h** Representative fluorescence image after perfusion with 40 kDa FITC-Dextran through the microvasculature for 20 min. Graph depicts fluorescence intensity across the microvasculature inside the dashed box at 7 min (red) and 10 min after perfusion (blue). Scale bar: 100 μm. **i** F-actin cytoskeleton arrangement in endothelial cells and related flow shear force. Right: enlarged view within the white boxes. Scale bar: 100 μm. **j** Representative confocal image showing new sprouting from the main vessel lumen after angiogenetic stimuli. Right image shows cross-sectional view (yz). CD31 (green), Vimentin (red), DAPI (blue). Scale bar: 100 μm. Data are provided in a Source data file.

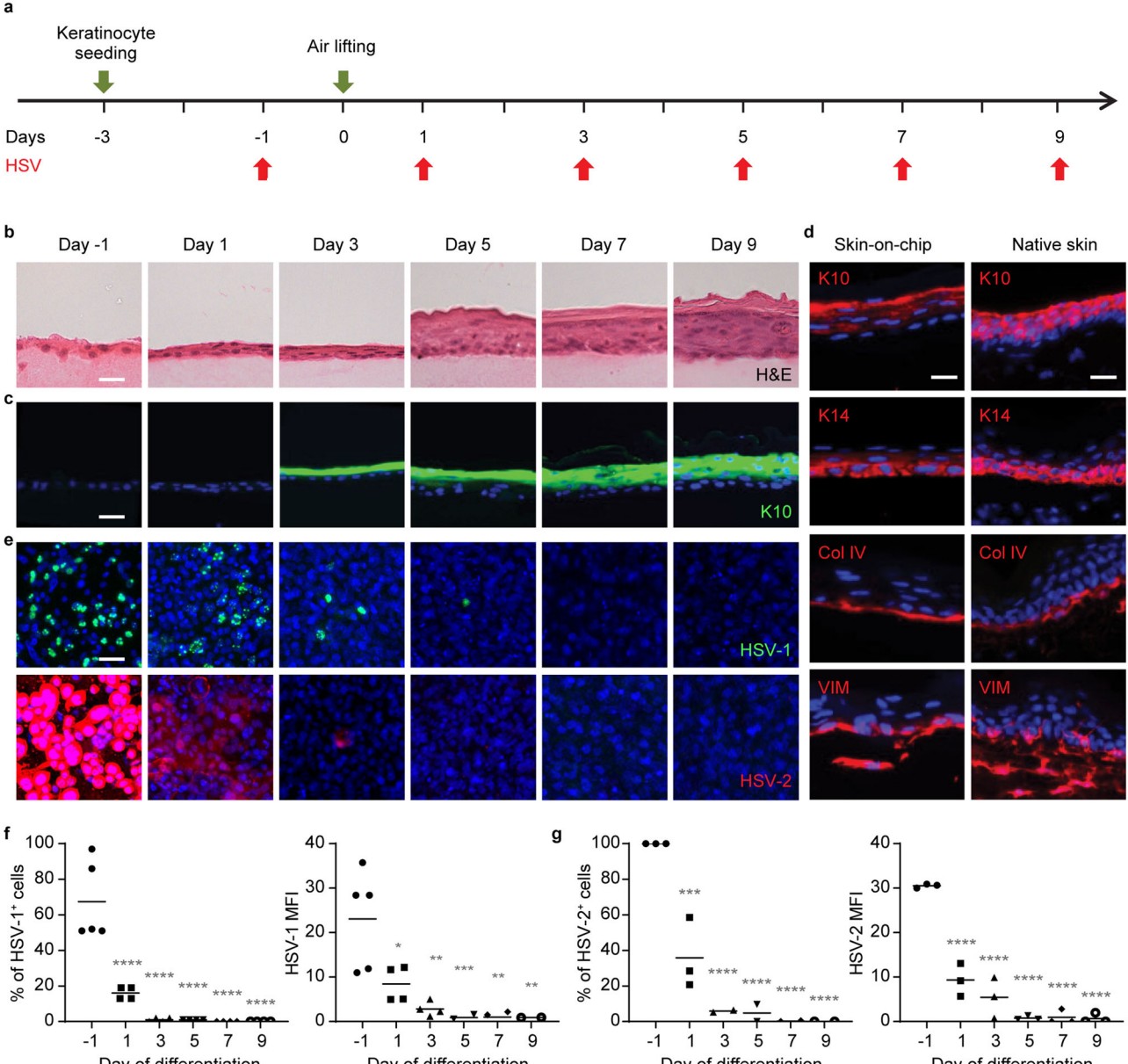

**Fig. 2 | HSV infectivity at different stages of keratinocyte differentiation.**
**a** Schematic of HSV epidermal infection during keratinocytes differentiation in the skin-on-chip. Keratinocytes were infected with HSV ($10^6$ PFU) 1 day prior to air lifting or 1, 3, 5, 7, or 9 days post air lifting and examined 24 h post infection.
**b** Representative cross-section H&E images of the skin-on-chip (mock infected) at different time points before and after air lifting. Scale bar: 20 μm. **c** Expression of keratinocyte differentiation marker Keratin 10 (K10) in mock infected skin-on-chip at times (indicated in **b**). K10 (green), DAPI (blue). Scale bars: 20 μm. **d** Expression of keratinocyte markers K10 and K14, basement membrane marker collagen IV (Col IV) and stromal marker vimentin (VIM) in the epidermis of skin-on-chip (left) compared with native human skin (right). Scale bars: 20 μm. **e** HSV infection in the epidermis of skin-on-chip by HSV-1 strain K26 (green, upper row) or by HSV-2 strain 186 (red, bottom row). DAPI (blue). Scale bar: 20 μm. **f, g** Infection was initiated at indicated keratincyte differentiation day and assessed 24 h later. Statistics generated by repeated measures one-way ANOVA with Tukey's post-test and indicates difference from −1 day. **f** Quantification of virally infected cells and mean fluorescence intensity (MFI, $*p = 0.0198$, $**p = 0.0012$, $***p = 0.0003$, $**p = 0.0012$, $**p = 0.004$) of viral gene expression for HSV-1 ($n = 5$). **g** Quantification of virally infected cells ($***p = 0.0006$) and MFI of viral gene expression for HSV-2 ($n = 3$). $****p < 0.0001$ for all graphs. Data are provided in a Source data file.

replication once the cornified layer was developed (Fig. 2e–g). In contrast, the undifferentiated keratinocyte monolayer (day −1) was most susceptible to HSV infection. Even 1-day post air lifting, the number of infected cells, on average, decreased dramatically from 67% to 16% for HSV-1 and from 100% to 36% for HSV-2 (Fig. 2f). Notably, the rough surface of the epithelium at day −1 became smooth and flattened one day following air lifting (Fig. 2b). At this time, a second keratinocyte layer appeared, likely the early stages of the suprabasal layer. The morphological changes shown by H&E and fluorescence staining suggest keratinocytes had initiated upward growth and differentiation,

even one day after air lifting. HSV infectivity diminished rapidly as keratinocytes differentiated further into K10+ layers. Virtually, no infected keratinocytes were detected in the epithelium after 5 days of differentiation. These results indicate the fully differentiated nature of this skin-on-chip epidermis and highlight the importance of intact skin barrier function in protection against HSV infection.

In order to simulate tissue microinjury that facilitates viral access to susceptible cells of the fully differentiated epidermis, we then mechanically disrupted the epidermis with a dermatology punch prior to the addition of virus (Fig. 3a). HSV infection mainly initiated in

keratinocytes surrounding the epidermal rupture and exhibited pathomorphological characteristics, including multi-nucleation formation, nucleus enlargement and chromatin margination of infected cells (Fig. b, c and Supplementary Fig. 6), similar to HSV ulcerations seen in human biopsy tissue (Fig. 3d). In the distal areas, K14+K10− basal keratinocytes were preferentially infected with HSV, further indicating the critical role of basal keratinocytes in acquiring and spreading HSV infection in skin (Fig. 3e). Together, these data demonstrate that the bioengineered 3-D skin-on-chip emulates native HSV infection in human skin, providing a feasible platform for modeling human HSV infection in vitro.

### Evaluation of inflammatory responses to HSV infection in the 3-D vascularized skin-on-chip

Functional blood vessels are capable of mediating immune-cell infiltration in response to tissue infection and environmental cues. We perfused neutrophils, freshly isolated from human peripheral blood, through the endothelialized microvascular network and examined their responses to HSV infection in the skin-on-chip (Fig. 4a and Supplementary Fig. 7). Neutrophils were monitored in real-time since perfusion for their trans-endothelial activities via live-cell imaging. Rapid attraction and adhesion of neutrophils to the vessel wall occurred within minutes of perfusion in the skin-on-chip infected with HSV, whereas minimal adhesion of neutrophils was observed in the mock infection (Fig. 4b). As infection continued, neutrophils accumulated along the microvessel wall, transmigrated through the endothelium, and entered the dermal space in response to HSV infection (Fig. 4c, d). Real-time imaging revealed

neutrophil trans-endothelial migration across the endothelium and rapid movement in the dermal space (Fig. 4d and Supplementary Movie 3). Thus, the endothelialized microvascular network demonstrated proper function in supporting inflammatory immune response to infected tissues.

We then further examined the spatial relationship between infiltrating neutrophils and infected epidermis in the skin-on-chip. Confocal images revealed that neutrophils migrated upward, against gravity, towards the infected epidermis, and formed close interactions with HSV-infected keratinocytes (Fig. 5a–c). Quantitative analysis confirmed that the majority of neutrophils had extravasated and migrated toward the epidermis, with a portion of them reaching the infected keratinocytes at 20 h after the initiation of perfusion (Fig. 5c). We analyzed the levels of IL-6, IL-8, RANTES, and TNFα secretion in the circulating media to understand the underlying mechanism of neutrophil chemoattraction. The monitoring of proinflammatory cytokine and chemokine production during HSV infection was achieved through the fluidic access points designed for each structural compartment in the 3-D skin-on-chip (Supplementary Figs. 1 and 2). HSV infection of epidermal keratinocytes markedly induced IL-8, IL-6 and to a much lesser degree RANTES production, but TNFα remained undetectable in both mock and HSV-infected skin-on-chip (Fig. 5d). IL-8 was the most abundantly expressed at baseline and in response to HSV infection among the three detectable cytokines. The predominance of IL-8 expression induced during HSV infection supported rapid recruitment and massive infiltration of neutrophils (Figs. 4b and 5a). To further investigate the role of IL-8 in chemoattraction and extravasation of neutrophils, we treated the skin-on-chip with a

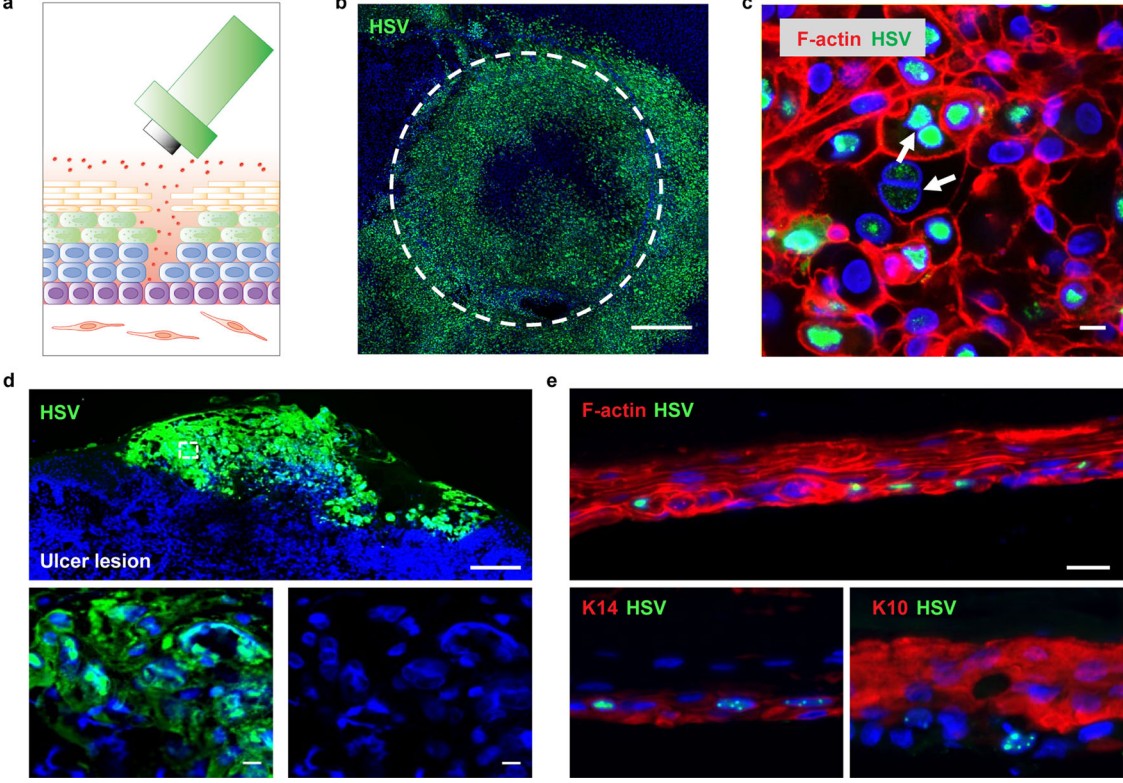

**Fig. 3 | HSV Infection in epidermis of human skin-on-chip and native human skin. a** Schematic of epidermal HSV infection protocol by a biopsy punch. HSV-1 K26 virus ($10^6$ PFU) was added on top of the epithelium after the stratified epidermis was disrupted by a 1.5 mm biopsy punch. **b** GFP expression of HSV-1 K26-infected 3-D skin-on-chip epidermis at 24 h post infection. Image shown as a maximum intensity projection of confocal z stacks. HSV-1 K26 (green). Scale bar: 500 μm. **c** HSV infection in the epidermis of skin-on-chip. Arrows indicate

margination of chromatin, enlargement of the cell nucleus, and multi-nucleation. F-actin (red), HSV-1 K26 (red), DAPI (blue). Scale bar: 20 μm. **d** Human ulcerative HSV lesion in a representative genital skin biopsy. HSV (green), DAPI (blue). Scale bar: 200 μm (upper) and 20 μm (lower). **e** HSV infection of the basal layer (indicated by K14 staining) of the 3-D skin-on-chip epidermis. HSV-1 K26 (green), DAPI (blue), K14 or K10 (red). Scale bar: 20 μm.

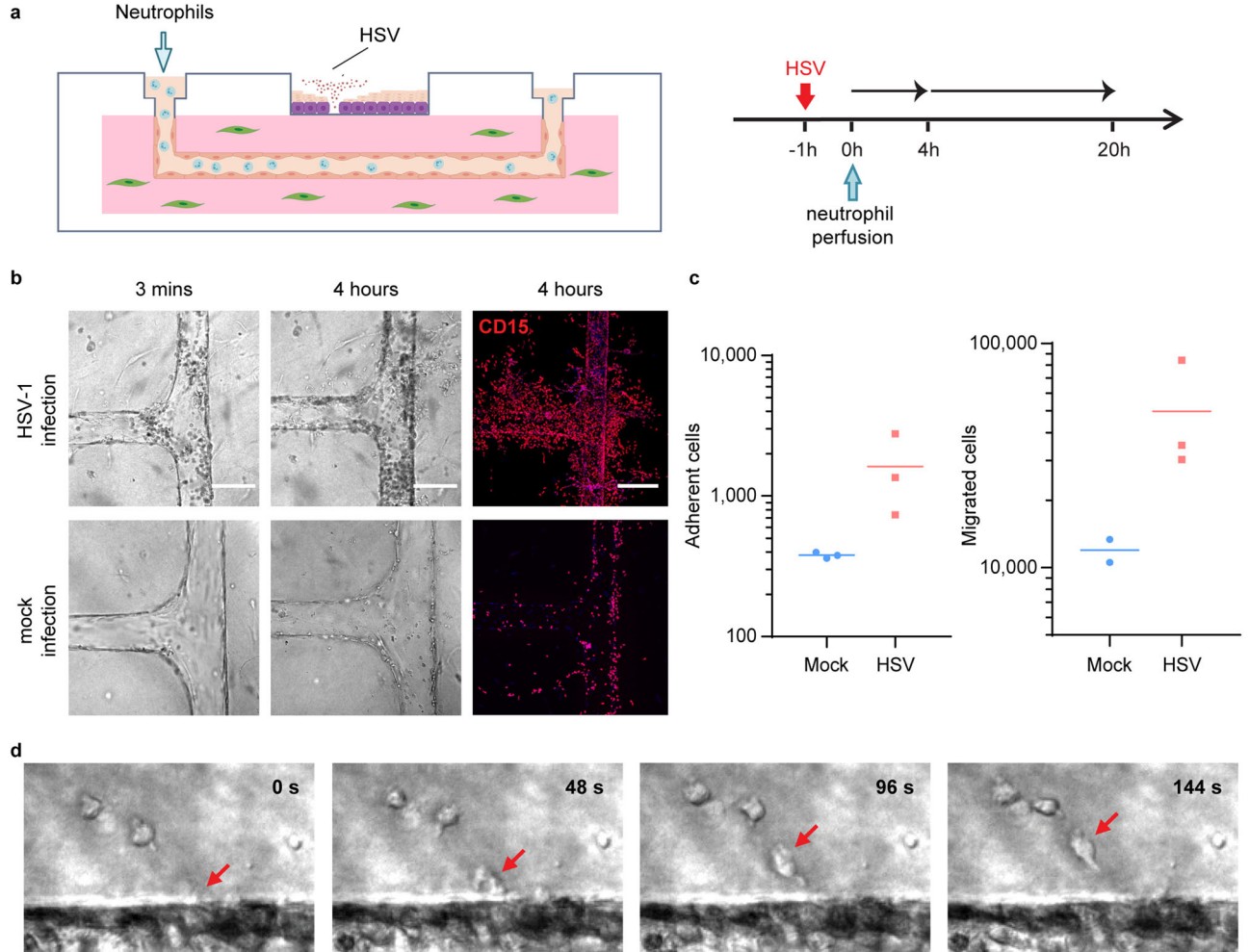

**Fig. 4 | Transmigration of neutrophils in HSV-infected skin-on-chip. a** Schematic of epidermal HSV infection and neutrophil addition to the skin-on-chip. Neutrophils were perfused through the microvascular network 1 h after the epidermis was infected with HSV-1 K26 (10⁶ PFU). The skin-on-chip was fixed either 4 h or 20 h post neutrophil perfusion. **b** Rapid neutrophil response to HSV infection in the epidermis. Bright field and maximum intensity projection confocal images of microvessels 3 min or 4 h post neutrophil perfusion in HSV-1-infected (top) or mock-infected (bottom) skin-on-chip. CD15 (red), DAPI (blue). Scale bar: 100 µm. **c** Neutrophil adherence (left) and transmigration (right) in mock- (blue) and HSV-infected (red) skin-on-chip 4 h post perfusion through the microvasculature. **d** Bright field images recording neutrophils transmigration across endothelium (arrows follow a single neutrophil). Data are provided in a Source data file.

neutralizing antibody specific for IL-8. Perfusion with an IL-8 neutralizing antibody resulted in a dramatic reduction of neutrophil vascular adhesion and diminished trans-endothelial migration (Fig. 5e, f). Blocking IL-8 function also changed neutrophil cytoskeleton dynamics, from elongated, pseudopodia extending transmigratory neutrophils to rounded cells that remained inside the vessels (Fig. 5e).

The directional migratory response by neutrophils was consistent with in vivo observations during ulcerative HSV reactivation in humans, where we observed neutrophils infiltrating toward the lesion site, engulfing infected cells (Fig. 6a). Microarray analysis of whole tissue level gene expression[45] confirmed significantly elevated levels of *IL8* and *IL6* gene expression in HSV lesions compared to uninvolved control tissue. Expression of *RANTES* and *TNFA* gene was also elevated in human lesion biopsies (Fig. 6b). Further, using fluorescence in situ hybridization (FISH) combined with immunohistochemistry (IHC) staining, we showed that keratinocytes expressed *IL8* mRNA in HSV-infected epidermis (Fig. 6c). In fact, both the K14⁺K10⁻ basal keratinocytes inside the herpetic lesion and the differentiated K14⁻K10⁺ keratinocytes adjacent to the ulcer expressed high level of *IL8* transcripts, but keratinocytes in unaffected normal human skin did not. These observations in HSV ulcer-bearing human skin support the notion that neutrophil chemoattraction and infiltration to HSV-infected skin might be dependent on functional IL-8 production by epidermal keratinocytes.

We also assessed expression of *RANTES* and *TNFA* transcripts in control and HSV lesion biopsies and showed positive detection in an HSV active lesion (Fig. 6d, e). In contrast to *IL8* transcript, which was predominantly expressed in keratinocytes, *RANTES* transcription expression appeared high in inflammatory infiltrates underneath the epidermis, but low in CD3⁺ T cells and K14⁺ basal keratinocytes (Fig. 6d). The scarce *RANTES* expression in keratinocytes is consistent with its low production seen in our skin-on-chip model (Fig. 5d). *TNFA* was found at low levels in the lesion epidermis and could be readily detected in CD3⁺ T cells (Fig. 6e). The data suggest that, upon HSV infection in skin, keratinocytes likely are the initial sources for proinflammatory cytokine production, such as IL-8. Other proinflammatory innate cells are potentially responsible for RANTES expression, contributing to subsequent T cell recruitment and host defense. Therefore, a perfusable, immune-competent skin-on-chip platform in combination with detailed in situ evaluation might provide a practical in vitro system to delineate complex cascades of immune activation in human HSV infection.

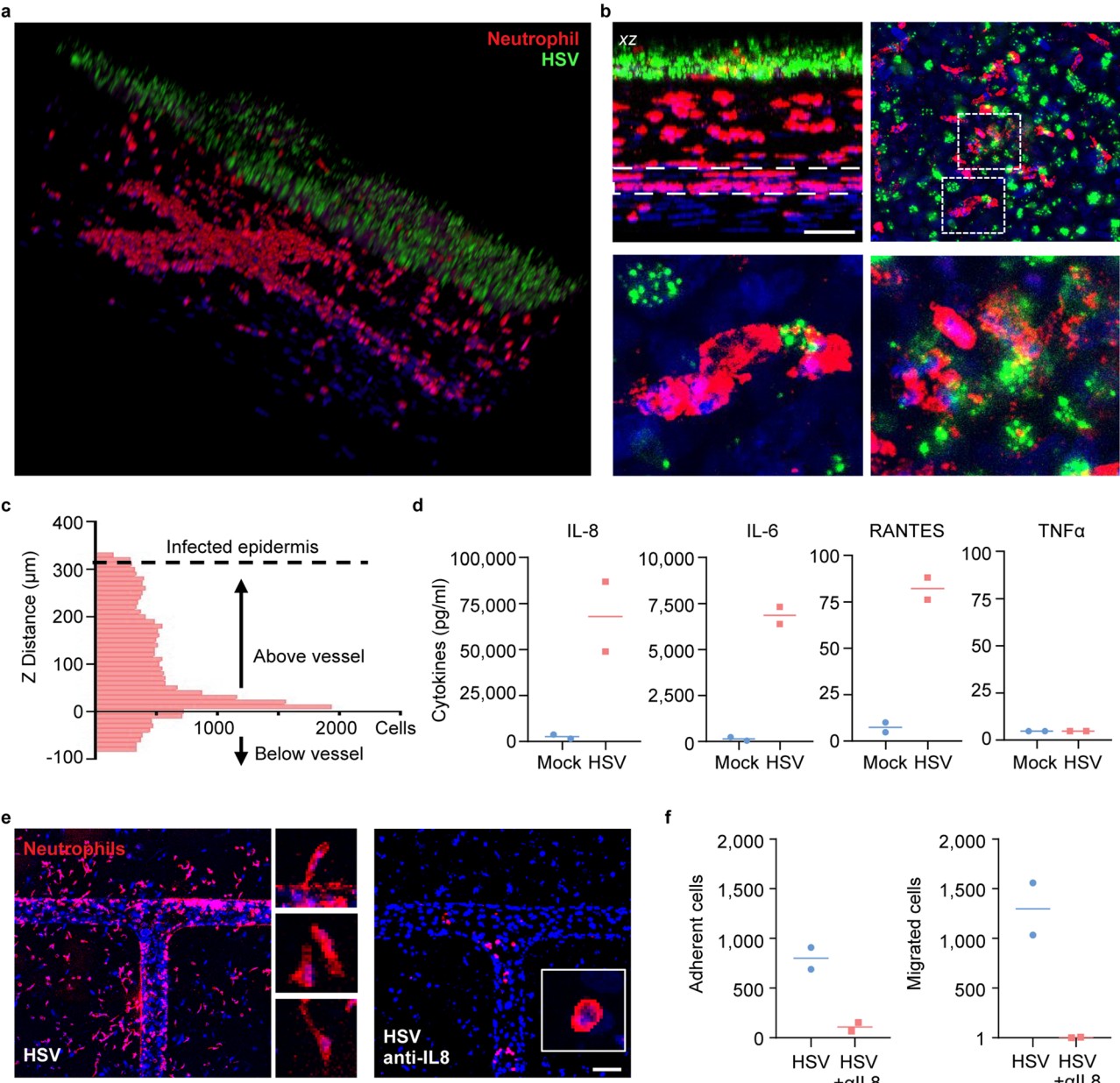

**Fig. 5 | IL8-dependent directional migration of neutrophils towards infected epidermis. a** Reconstructed 3-D confocal image showing the spatial distribution of infiltrating neutrophils (red, CD15) and HSV-infected epidermis (green) in the skin-on-chip. Neutrophils were perfused through the microvascular network 1 h after the epidermis was infected with HSV-1 K26 ($10^6$ PFU). Scale bar: 100 μm. **b** Top left: Projected cross-sectional view of neutrophil (red) migration into the dermal space toward HSV-infected (green) epidermis in the skin-on-chip. Dashed line indicates the vessel wall. Top right: Horizontal view of infiltrating neutrophils in the HSV-1-infected epidermis. Bottom row: enlarged views of the dashed boxes showing close contacts between neutrophils and infected keratinocytes. **c** Quantification of neutrophil distribution in z distance. Z = 0: neutrophils inside the vessel; Z > 0:

neutrophils migrated out of vessel and moved up toward the infected epidermis. Z < 0: neutrophils migrated out of vessel and moved down, away from epidermis. **d** Quantification of IL-8, IL-6, RANTES, and TNFα protein expression in culture medium after 5 h of mock or HSV-1 infection (*n* = 2). **e** Neutrophil extravasation in response to HSV infection in skin-on-chip either mock-treated (left) or treated with anti-IL-8 neutralizing antibody (right) for 6 h prior to infection. Neutrophils were perfused through the microvascular network 1 h after the epidermis was infected with HSV-1 K26 ($10^5$ PFU). Images taken 20 h post infection. Inset images show neutrophil morphology. CD15 (red), DAPI (blue). Scale bar: 100 μm. **f** Quantification of neutrophils that adhered to the vessel wall or transmigrated out of the vessel (*n* = 2). Data are provided in a Source data file.

## Perfusion of antiviral drug in 3-D skin-on-chip

We perfused the antiviral drug acyclovir (ACV) through the microvascular network, mimicking the in vivo route of oral and intravenous drug delivery, in order to assess the impact of dosage and timing on antiviral efficacy (Fig. 7a–c). Perfusion of ACV resulted in a dose-dependent suppression of HSV-2 infection (Fig. 7d and Supplementary Fig. 8). Treatment with 20 μM ACV, a dosage close to the serum concentration of standard treatment[46], resulted in a 78% reduction in the percentage of GFP-expressing cells and an 82% reduction in mean

fluorescent intensity, in comparison to a mock-treated condition (Fig. 7e). Next, we perfused ACV (20 μM) 24 h prior to, immediately upon or 24 h post infection (hpi) and examined the GFP expression of K26 after 48 h of infection (Fig. 7f). We found that the timing of treatment significantly impacted ACV antiviral efficacy. Treatment of ACV 24 h prior to infection or immediately upon infection significantly improved antiviral activity compared to perfusion of ACV 24 hpi (Fig. 7g–i). The potency of early antiviral treatment is reflected by a dramatic reduction in infected cell numbers and GFP expression

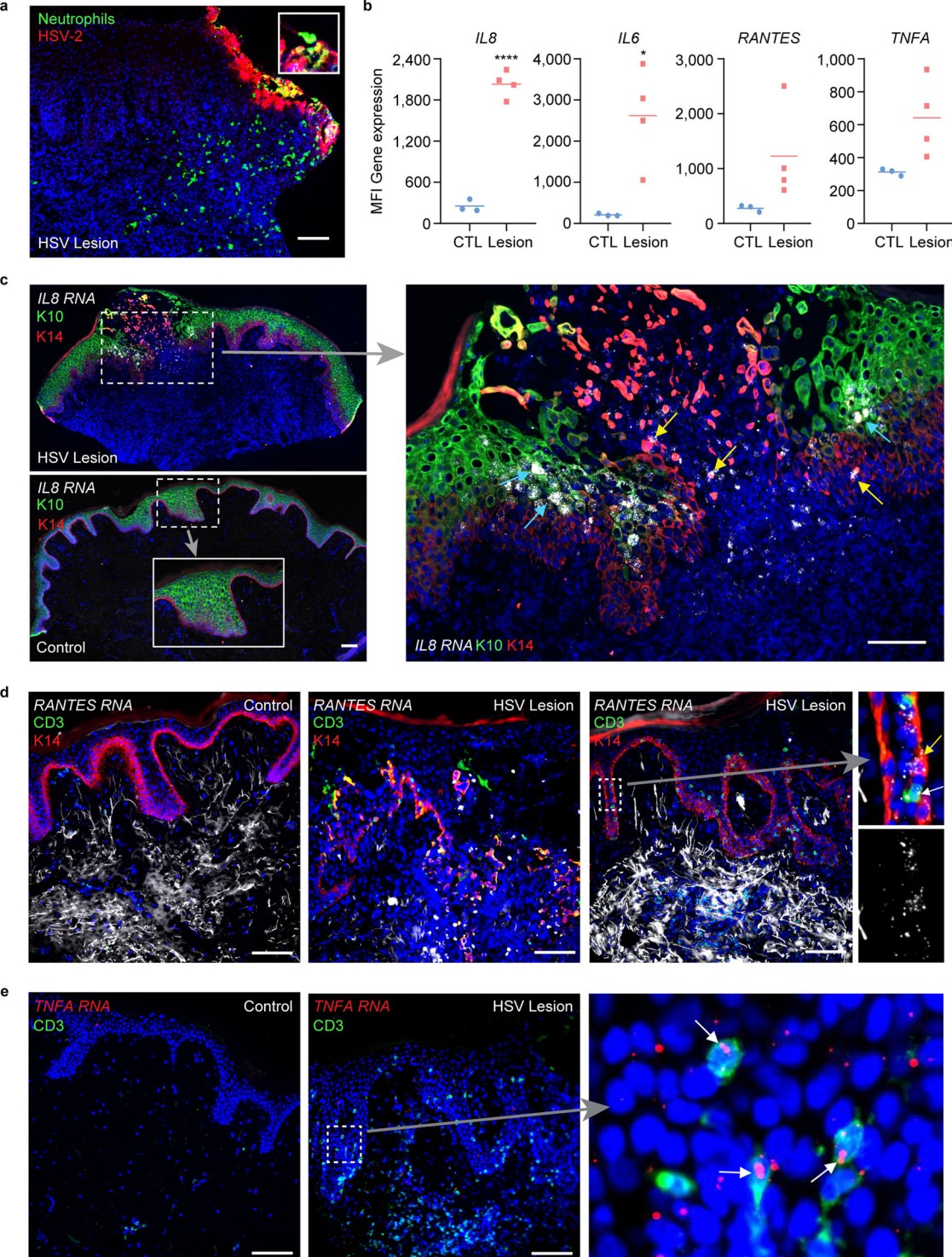

**Fig. 6 | Neutrophil migration and cytokine expression in human genital HSV ulcers. a** Neutrophil infiltration from dermis into HSV-infected epidermis in a human skin biopsy representative of an ulcerative herpes lesion. CD15[+] neutrophils (green), HSV (red). Scale bar: 100 μm. **b** Transcriptional levels of *IL8*, *IL6*, *RANTES*, and *TNFA* in human skin biopsies obtained from sites of HSV ulcerative lesion (*n* = 4) or from uninvolved contralateral controls (CTL, *n* = 3). Microarray data re-analyzed from a published data source[45]. *p = 0.0185 ***p < 0.0001 generated with paired two-tailed *t*-test. **c** Detection of epidermal *IL8* RNA transcripts using fluorescence in situ hybridization (FISH) in human skin biopsies. HSV lesion (upper) and matched normal control at contralateral site (bottom). Enlarged image of dashed box in HSV lesion showing *IL8* expression in K14[+] basal keratinocytes (red, indicated by yellow arrows) and in K10[+] differentiated keratinocytes (green, indicated by white arrows) (right). Scale bar: 100 μm. **d** *RANTES* RNA transcripts expression by FISH in control (left) and lesion (center and right) biopsies. Enlarged image of dashed box showing combined *RANTES* expression in K14[+] keratinocytes and CD3[+] T cells (top) and of only *RANTES* expression (bottom). K14 (red), CD3 (green), *RANTES* (white), DAPI (blue). Scale bar: 100 μm. **e** *TNFA* RNA transcripts expression by FISH in control (left) and lesion (right) biopsies. Enlarged image of dashed box (right). CD3 (green), TNFA (red), DAPI (blue). Scale bar: 100 μm. Data are provided in a Source data file.

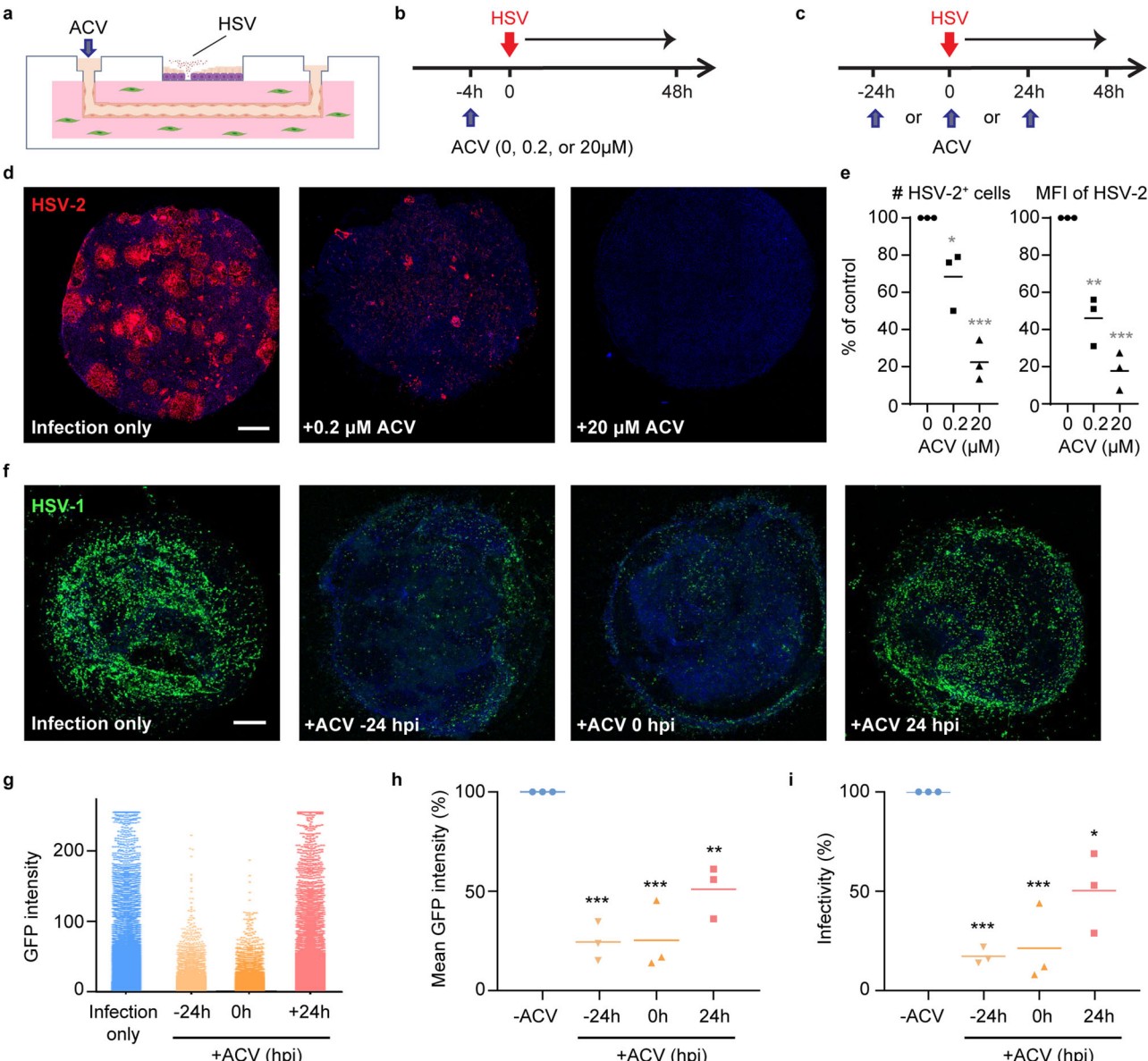

**Fig. 7 | Perfusion of acyclovir (ACV) through microvascular network inhibits epidermal HSV infection. a** Schematic of HSV epidermal infection and antiviral drug ACV perfusion. **b** ACV perfusion through the microvasculature at the concentrations of 0, 0.2, or 20 μM, 4 h before the epidermis was infected with HSV-2 ($10^6$ PFU). Skin-on-chip was fixed and stained for HSV gene expression 48 h post infection. **c** ACV perfusion (20 μM) at time points either 24 h prior HSV infection, immediately following infection or 24 h after the epidermis was infected with HSV-1 K26 ($10^6$ PFU). Skin-on-chip was fixed and stained 48 h after infection. **d** HSV infection in the epidermis under ACV perfusion at different concentrations as described in (**b**). Maximum intensity projection of stacked confocal images. HSV-2 (red), DAPI (blue). Scale bar: 500 μm. **e** Number of infected cells (*$p = 0.0298$, ***$p = 0.0003$) and MFI (**$p = 0.0011$, ***$p = 0.0001$) of HSV gene expression in the infected epidermis. Data normalized to untreated control across ACV conditions. $n = 3$. **f** HSV infection in the epidermis under ACV perfusion at different time points as described in (**c**). Maximum intensity projection of stacked confocal images. HSV-1 K26 (green), DAPI (blue). Scale bar: 500 μm. **g** Levels of GFP expression in epidermal cells infected with HSV-1 K26 with or without ACV perfusion as described in (**c**). Each dot represents one cell. **h** Mean GFP fluorescence intensity following HSV infection under various ACV treatment conditions relative to untreated control, $n = 3$. ***$p = 0.003$, **$p = 0.0046$. **i** Percentage of GFP+ cells in each condition, data were normalized to untreated control, $n = 3$. ***$p = 0.0005$, ***$p = 0.0007$, *$p = 0.0117$. Statistics generated by repeated measures one-way ANOVA with Tukey's post-test (indicates difference from control). Data are provided in a Source data file.

intensity of infected cells, which highlights the importance of early drug administration. No difference was observed between pretreatment and immediate treatment likely because ACV interference with viral DNA synthesis depends on viral thymidine kinase gene expression[47]. These experiments demonstrate that the endothelialized microvascular network was competent to drug perfusion in a dose-dependent and time-sensitive manner. Therefore, our 3-D skin-on-chip microfluidic platform might be used for preclinical drug efficacy testing and determining pharmacokinetics (PK)/pharmacodynamics (PD) of potential therapeutics.

## Discussion

We have bioengineered a microfluidic-based, full-thickness skin-on-chip infection model that features circulating immune cells and antiviral agents with an endothelialized microvessel network. This vascularized 3-D skin-on-chip system recapitulates key structural and

functional properties of normal human skin, including blood vessel-like microvasculature with the capacity to mediate nutrient uptake, inflammatory immune responses, and therapeutic drug delivery. HSV infection of the stratum epidermis in the skin-on-chip results in pathophysiological effects mimicking HSV ulceration in humans. Further, the system is permissible to real-time monitoring and quantitative assessment of human immune responses during pathogen invasion. The proof-of-concept application of an antiviral drug demonstrates its potential for use as an effective and cost-efficient preclinical drug-testing pathway. Thus, this interactive and dynamic in vitro human skin model might be a promising platform, alternative to animal models, for skin disease research and therapeutic intervention.

The design and innovative fabrication and construction of the microfluidic skin-on-chip enable incorporation of an endothelialized vasculature in the dermis and allow for monitoring host immune response to viral infection in vitro. Our model comprises several distinctive features. First, the system contains a dermis with stiffer collagen (7.5 mg/ml) compared to that of traditional skin equivalents (<3 mg/ml)[48,49] which often suffer from substantial contraction and shrinkage induced by fibroblast cells over time. In addition, we coated the device jig with polyethyleneimine and glutaraldehyde to enhance the adherence of collagen, further reducing collagen shrinkage and de-attachment from the jig even weeks after culturing. These improvements minimized changes in the collagen matrix over time, which allowed seeding of endothelial cells that were able to maintain the designed geometry of a microvascular network. Second, we created individual access ports to the epidermis, dermis, and microvessels, granting conditional medium delivery and treatment to each compartment without affecting others. Our microfluidic skin-on-chip allowed continuous gravity-driven perfusion through this vascular network, providing necessary mechanical signals, supply of nutrients, and drainage of metabolites, which has been reported to enhance epidermal morphology and barrier function[50]. Furthermore, the epidermis, dermis, and perfusable vasculature in the skin-on-chip were in direct contact with collagen and other extracellular matrix components, further promoting cell–cell and cell-matrix crosstalk among keratinocytes, fibroblasts, endothelial cells, and the surrounding microenvironment. We have designed the skin-on-chip to have a total height of 500 μm (from bottom coverslip to top epidermis), which is suitable for both live cell imaging and in situ visualization by confocal microscopy. Thus, the extravasation and migration of immune cells could be observed, characterized, and quantitatively assessed. Most importantly, our vascularized skin-on-chip is free of poly-dimethylsiloxane (PDMS), which is a widely used material in microfluidic systems[51] but has the well-known propensity to absorb small molecules and interfere with drug delivery. Taken together, these features provide our skin-on-chip a more physiological environment for in vitro study of human skin in health and disease.

Our 3-D skin-on-chip model also made it possible to investigate HSV infection during keratinocyte differentiation. We confirmed that basal keratinocytes in the stratum basale yielded the highest viral infectivity and were the most susceptible targets of HSV infection. As keratinocytes differentiated, they seemed to be less susceptible to HSV. So far, transcriptional profiling analysis provided no evidence of enhanced intrinsic antiviral states for viral resistance upon keratinocytes differentiation[14,52]. The decrease in infectivity during keratinocyte differentiation and stratification processes might be due to reorganization of adhesive molecules to form intercellular tight junctions, thereby reducing accessibility of cellular receptors and rendering resistance to virus spread and transmission[53,54]. Interestingly, human biopsy studies have shown that skin-innervating sensory nerve endings, which release reactivated virus particles into skin and mucosal surface, extend neurites directly to the basal keratinocytes at the dermal-epidermal junction (DEJ)[10,11]. In the same dynamic spaces,

CD8+ T cells, including HSV-specific CD8+ T cells, persist and patrol at the DEJ, forming close contacts with basal keratinocytes to prevent virus dissemination[12,14]. Thus, both in vitro and in vivo data support the notion that basal keratinocytes are the primary targets in the battle between reactivated virus and protective host immune defense. Our data also corroborate the importance of skin barrier function in preventing HSV acquisition.

HSV infection induces various proinflammatory cytokines and chemokines in epithelial cells[55–57]. In the 3-D skin-on-chip system, HSV infection of epidermal keratinocytes resulted in secretion of proinflammatory cytokines and chemokines, IL-8, IL-6, and to a lesser degree RANTES. IL-8, the most highly detected in our skin-on-chip, is a key chemoattractant for neutrophils in humans[58,59], but absent in the genomes of rodents[60]. Thus, a human model system is required for studying the role of IL-8 in neutrophil activation, migration, and function. Our vascularized skin-on-chip device permitted both live cell imaging and in situ characterization by confocal microscopy to observe and quantify neutrophil extravasation and migration. We have shown that the free-flowing neutrophils slowed, rolled along the vessel wall, and migrated through the endothelial layer in response to HSV infection. When perfused with neutralizing antibody specific to IL-8 through the vascular network, neutrophil trans-endothelial migration was inhibited effectively. The early migration of neutrophils and predominant production of IL-8 in our skin-on-chip HSV infection model are consistent with the in vivo evidence of neutrophils being the first responders to HSV infection in humans[61–63]. Furthermore, detailed in situ characterization suggested that differentiated keratinocytes could be the major sources of initial inflammatory cytokine production upon HSV infection in skin. This finding might explain the lack of IL-8 detection in conventional keratinocyte cultures and underline the importance of implementing 3-D organoid or organ-on-chip platforms over the 2-D tissue culture system in biomedical research.

One of the most significant applications for organ-on-chip systems is in the field of drug discovery and validation. We have shown that microvascular perfusion of ACV in the dermal compartment, which mimics oral intake and intravenous injection routes of drug delivery, inhibits HSV replication in the epidermis layer of the skin-on-chip in a dose-dependent and time-sensitive manner. Perfusion with 20 μM ACV, which is comparable to the peak serum concentration (27 μM) when taking the recommended daily oral dosage of 1000 mg valacyclovir[46], achieved ~80% inhibition of HSV-2 infection in our model. The data also indicate the necessity of early treatment for effective viral clearance, which agrees with clinical practice[64,65]. The effectiveness of early treatment helps explain clinical findings where long-term daily oral administration of ACV suppresses genital herpes in patients who have frequent recurrences, while episodic treatment is not efficacious in most of these patients[66]. These results suggest that the vascularized 3-D skin-on-chip system would afford a heretofore unachieved fidelity in an in vitro drug testing setting.

The limitations of our skin-on-chip device include the utilization of mixed origins and commercially available sources of primary human skin cells and lack of skin-resident immune components. Although our device is suitable and relevant for innate immune evaluation and antiviral drug testing, investigation of T and B cell-mediated adaptive immune responses to infection requires matched HLA alleles of immune components and tissue structure cells. The fact that little TNFα expression was detected in the current model of HSV infection indicates the limitations of compositing only structural components in skin-on-chip devices. Tissue-resident immune cells are to be incorporated as they likely play an important role in initial pathogen sensing and subsequent cytokine/chemokine induction to coordinate T cell recruitment and host defense. While our current platform provides a useful in vitro system, the next generation of the 3-D skin-on-chip should contain autologous cellular components, whereby keratinocytes, fibroblasts, and dermal endothelial cells are isolated primarily

from a single donor or are differentiated utilizing regeneration technologies from stem cells, to delineate complex in vivo human immune responses to HSV infection. In the time ahead where precision medicine becomes increasingly important, personalized skin-on-chips might be essential for characterizing immune mechanisms underlying a wide-spectrum of disease outcomes, such as those presented in genital herpes infection, as well as for performing preclinical evaluation of appropriate treatments, including therapeutic drugs and immune therapies.

Taken together, the 3-D skin-on-chip system incorporates key elements for modeling skin diseases to decipher host immune interactions in vitro, including intact skin architecture of the epidermis and dermis, functional endothelial vessels, infectious pathogens, immune components, and therapeutic agents. The skin-on-chip is highly promising in mimicking human HSV infection, immune responses, and antiviral treatment. This platform offers a powerful complement to animal models for gaining fundamental insights into human disease and therapy.

## Methods

### Ethical statement
The study protocol involving human specimens was approved by the University of Washington Institutional Review Board Committee (STUDY00002443), and written informed consents were obtained from all participants. Study participants were HSV-2 seropositive and HIV-1 seronegative healthy individuals ($n = 20$), who had a history of culture proven recurrent genital HSV-2 disease. Both male ($n = 4$) and female ($n = 16$) participants, ages between 20 and 71 years old (average at 48.4), were included. Three-millimeter punch biopsies were obtained during clinical symptomatic recurrences from active genital herpes lesion sites as described previously[10,14,45]. Control skin biopsies were taken from normal unaffected contralateral area. All tissue samples were fresh frozen in optimum cutting temperature compound (OCT) and stored at −80 °C until processing.

### Collagen gel preparation
Type 1 collagen was extracted from rat tail tendon following the standard protocol[67]. Collagen fibers of the rat tail tendons were quickly separated, collected, and rinsed three times with 70% ethanol. The fibers were then air dried and dissolved in 0.1% acetic acid (Sigma) for a minimum of 48 h at 4 °C. After being solubilized, collagen fibers were centrifuged, and the supernatant was collected, frozen, lyophilized, and resuspended with 0.1% acetic acid at a concentration of 15 mg/ml. The stock collagen gel was stored at 4 °C.

### Generation of human dermal fibroblast-containing collagen
Human primary dermal fibroblasts (Lifeline Cell Technology, FC-0001) were cultured in Fibrolife serum-free medium and maintained at 37 °C in a humidified incubator under 5% $CO_2$ according to the manufacturer's protocols. Stock collagen was diluted and neutralized into 7.5 mg/ml with 10X M199 (Sigma, M2520), 1X FibroLife serum-free medium (Lifeline Cell Technology, LL-0001), and 1 N NaOH. Human primary dermal fibroblasts, before passage 5, were trypsinized, resuspended at $10^6$ cells/ml, and then 1 ml was mixed with 7.5 mg/ml collagen on ice.

### Fabrication of microfluidic network
The fabrication of a microvascular network was modified and adapted from an injection molding protocol previously developed by Zheng et al.[41]. The microvascular network was enclosed within a fibroblast-containing collagen construct and housed between two plexiglass pieces. The top plexiglass contained a well with two injection ports for injecting collagen into the top well, two ports as inlet and outlet for perfusing endothelial cells and medium, and one open well for keratinocyte culture. A PDMS stamp with channel geometry that was fabricated by photolithography and soft lithography was placed at the bottom of the top plexiglass jig and fibroblast-containing collagen was injected through the inject ports into the top well. Two dowel pins were inserted into perfusion ports to define the space for inlet and outlet, and a flat piece of stainless steel was placed on top of the open well to create a flat collagen surface for keratinocyte culture. The bottom plexiglass also provided a well for collagen. After fibroblast-containing collagen was added into the well, a flat PDMS stamp was placed on top to create a flat collagen gel. Collagen in both top and bottom plexiglass was incubated at 37 °C for 25 min for gelation. Later, PDMS stamps were removed, and top and bottom plexiglass jigs were assembled and sealed by mechanical pressure. Fibroblast culture medium FibroLife (Lifeline Cell Technology, LL-0001) was perfused through the inlet port and the assembled device was incubated at 37 °C in a humidified incubator under 5% $CO_2$.

### Generation of endothelialized microfluidic network and perfusion culture
Human primary dermal microvascular endothelial cells (Lonza, CC-2516) were cultured in EGM-2 MV medium (Lonza, CC-3162) according to manufacturer's protocol and used before passage 6. The cells were maintained at 37 °C in a humidified incubator under 5% $CO_2$. Endothelial cells were added one day after the generation of the microfluidic network to allow for embedded fibroblasts to settle. Before seeding endothelial cells, the device was perfused with EGM-2 MV medium for 1 h 10 μl of human dermal microvascular endothelial cells were seeded through the inlet port at about $5 \times 10^6$ cells/ml and allowed to attach to the microvascular channel wall for 20 min. EGM-2 MV medium was added in the inlet and perfused through the microvascular channel under gravity force. Fresh medium was changed every 12 h to keep the gravitational flow between inlet and outlet. The microvascular network was perfused for at least 2 weeks before infection experiments.

### Generation of stratified epidermis on top of microvascular network
Human primary epidermal keratinocytes (Lifeline Cell Technology, FC-0025) were cultured and proliferated in growth medium (CELLnTEC, CnT-PR) and only early passage cells were used for differentiation. The cells were maintained at 37 °C in a humidified incubator under 5% $CO_2$. Ten thousand epidermal keratinocytes were seeded on the top well of the device and cultured within CnT-PR medium for 24 h. More epithelial CnT-PR medium was also added in the injection port. Simultaneously, the endothelial cell EGM-2 MV medium was constantly perfused through the channel to support endothelial cells. When keratinocytes reached 100% confluency, medium in the top well and injection port was changed into keratinocyte differentiation medium (CELLnTEC, CnT-PR-3-D). After overnight culture, medium in the top well was removed, while the differentiation medium in the injection port was maintained to provide nutrients for fibroblasts in collagen and keratinocytes on top of the collagen. Keratinocytes were then exposed to the air and allowed to differentiate for 7 to 12 days.

### Measuring vessel permeability
The quantitative assessment of endothelial permeability was performed and calculated as described previously[44,68]. Briefly, 10 μM FITC-Dextran (40 kDa, Sigma) was perfused at a continuous flow rate of approximately 5 μl/min through the microvascular network for 20 min. Live cell fluorescent images were then taken by a Nikon Ti microscope at every image per second. Determination of the permeability coefficient was based on the time lapse images of perfusion. A window ~550 μm long and 300 μm wide was drawn across the microchannel, and mean fluorescence intensity in the measuring window was

measured using ImageJ and plotted against time. Compute apparent permeability coefficient $P_a$ as follows:

$$P_a = \frac{1}{I_1 - I_b} \left( \frac{I_2 - I_1}{t_2 - t_1} \right) \frac{d}{4}$$

where $I_b$ is the background intensity, $I_2$ is the intensity at $t_2$, $I_1$ is the intensity at $t_1$ and $d$ is the width of the channel. Four small vessels were randomly chosen to obtain an averaged value for apparent permeability coefficient.

## In situ immunofluorescence staining and confocal imaging of skin-on-chip

The fluorescence imaging was performed without disassembling the skin chip. At the designated time point, the device was fixed in situ by perfusing 4% paraformaldehyde through the device and adding paraformaldehyde in the top well and ports at room temperature for 30 min. After three 15 min washes with PBS, the device was blocked and permeabilized with 2% bovine serum albumin and 0.5% Triton X-100 for 1 h. The device was then incubated with primary antibodies at 4 °C overnight. After three 15 min PBS washes, the device was incubated with secondary antibodies for 2 h. DAPI was used for nuclei counterstain. After three PBS washes to remove unbound antibodies, the device was ready for confocal imaging and analysis. The z-stacks of horizontal fluorescence images were required using a Zeiss LSM 780 NLO confocal microscope. Images providing an overview of the entire device were taken using a ×10 objective lens at a z-step of 10 μm. For a detailed view, images were taken using a ×40 objective lens at a z-step of 1 μm. The 3-D reconstitution and vertical cross-sectional view was generated from the z-stacks of images using ImageJ (version 2.3.0) and Volocity (version 6.5.0, Improvision). The primary antibodies used were (summarized in Supplementary Data 1): rabbit anti-human CD31 (1:50, Abcam, ab32457), mouse anti-human CD15 (1:50, BD Biosciences, 555400), rabbit anti-human keratin 14 (1:25, Abcam, ab192056), rabbit anti-human VE-cadherin (1:50, Abcam, ab33168), rabbit anti-human HSV-2 (1:500, Dako, B0116), and rabbit anti-human vimentin (1:50, Abcam ab92547). Secondary antibodies used were Alexa Fluor 594 donkey anti-mouse IgG antibody (1:100, Invitrogen A21203), Alexa Fluor 546 donkey anti-rabbit IgG antibody (1:100, Invitrogen A10040), Alexa Fluor 488 goat anti-mouse IgG antibody (1:100, Invitrogen A11001) and Alexa Fluor 647 goat anti-rabbit IgG antibody (1:100, Invitrogen A21244). F-actin were stained with Alexa Fluor 594 phalloidin (1:1000, Thermo Fisher Scientific, A12381).

## Immunofluorescence staining of skin-on-chip cryosections

The devices were fixed in situ by 3.7% paraformaldehyde at room temperature for 30 min and disassembled. The epidermis at the top was taken out and frozen in OCT and stored at −80 °C until processing. Frozen epidermis or human biopsy tissue was cryo-sectioned into 8 μm slices and air dried overnight. The slices were blocked and permeabilized with 2% bovine serum albumin and 0.5% Triton X-100. After 1-h incubation, the slices were stained overnight at 4 °C with primary antibodies and secondary antibodies for 1 h at room temperature. The primary antibodies for validating stratums (summarized in Supplementary Data 1): rabbit anti-human keratin 14 (1:25, Abcam, ab192056), rabbit anti-human keratin 10 (1:25, Abcam, ab76318), mouse anti-type IV collagen (1:100, Sigma-Aldrich, C1926), rabbit anti-human vimentin (1:100, Abcam ab92547), mouse anti-human filaggrin (1:50, Invitrogen, MA5-13440), rabbit anti-human loricrin (1:50, Abcam, ab85679), mouse anti-human involucrin (1:50, Invitrogen, MA5-11803), rabbit anti-human Ki67 (1:50, Abcam, ab15580), rabbit anti-human HSV-2 (1:500, Dako, B0116). Secondary antibodies used were Alexa Fluor 488 goat anti-mouse IgG antibody (1:100, Invitrogen, A11001) and Alexa Fluor 647 goat anti-rabbit IgG antibody (1:100, Invitrogen, A21244). After staining, the slices were mounted in Prolong Gold Antifade Mountant (Thermo Fisher Scientific, P36930) and imaged by Nikon Ti microscope equipped with a CCD camera.

## Combined fluorescence in situ hybridization (FISH) and immunohistochemistry (IHC) staining

Human skin biopsy samples were cryo-sectioned into 10 μm, mounted on slides, fixed with 4% paraformaldehyde, and dehydrated in ethanol. Dehydrated slides were pretreated with hydrogen peroxide and 0.3% Triton before FISH probe hybridization using RNAscope® Multiplex Fluorescent Reagent Kit v2 (Advanced Cell Diagnostics, 323100), according to manufactory's protocol. FISH probes: IL8 (Advanced Cell Diagnostics, 310381_C3), RANTES (Advanced Cell Diagnostics, 549171), and TNFA (Advanced Cell Diagnostics, 310421-C2). After probe hybridization, slides were washed with washing buffer (Advanced Cell Diagnostics, 323100) before continuing with the IHC staining. Slides were incubated with blocking buffer for 60 min at room temperature (RT) and then with primary antibodies overnight at 4 °C. Primary antibodies: Cytokeratin10-AF488 (1:25, Abcam, ab194229) and Cytokeratin14-AF647 (1:25, Abcam, ab192056). After washing with PBS with 0.1% Tween-20, slides were counterstained with DAPI (Themo-Fisher Scientific, D3571) and mounted in Prolong Gold Antifade Mountant (Thermo Fisher Scientific, P36930).

## HSV infection of skin-on-chip

Viral stocks used in this study include HSV-1 virus K26, which encodes the capsid protein, VP26, fused to GFP (a courtesy gift from Dr. Prashant Desai, Johns Hopkins University, Baltimore, MD), and HSV-2 strain 186. Viral titers were determined by titration in Vero cells (purchased from ATCC, cat# CCL-81). After a multilayered epidermis was developed, we applied mechanical force through a 1.5 mm biopsy punch to generate an incision on the epidermis and added $10^6$ plaque-forming units (PFUs) of virus on top of the epidermis. ACV was added at different time points in the inlet and perfused gravitationally through the microvascular channel. Mock infection and mock treatment were performed with media only. Fresh medium and ACV were perfused twice a day. Forty-eight hours after infection, skin chips were fixed with 4% paraformaldehyde, stained, and analyzed with a confocal microscope.

## HSV detection in biopsy tissue and skin-on-chip

We performed immunofluorescence staining using rabbit polyclonal antibody to detect HSV-2 antigen (1:500, Dako, B0116) on either OCT-embedded human skin lesion biopsies or HSV-infected skin-on-chip epidermis specimen. In addition, we used a highly sensitive PCR assay to detect HSV-2 genome copies for virus quantification in tissue[69].

## Perfusion of human neutrophils through the microvascular network in skin-on-chip

Neutrophils were isolated from fresh human whole blood as previously described[70]. Briefly, whole blood from a healthy donor was collected in an acid citrate dextrose tube. After dilution with HBSS (no calcium, no magnesium) (Gibco), the blood mixture was layered onto Ficoll-Paque Premium (GE healthcare life sciences) and centrifuged at $850 \times g$ for 20 min with brake off. The neutrophil- and erythrocyte-rich layer at the bottom was mixed with 3% Dextran (MP Biomedicals) and sat for 20 min at room temperature allowing erythrocytes to sediment under gravity. The neutrophil-rich supernatant was transferred into a fresh tube and centrifuged at $300 \times g$ for 10 min. The cell pellet was incubated with ACK lysing buffer (Gibco) for 5 min to lyse the remaining erythrocytes. The cells were then washed with HBSS twice and resuspended in RPMI 1640 with 10% fetal bovine serum and 1X penicillin streptomycin (Gibco). The purity of isolated neutrophils was checked via flow cytometry for each experiment. A fraction of the cells was stained with an optimized antibody cocktail for 20 min at room temperature, washed with PBS containing 2% FBS and fixed in PBS containing 1% paraformaldehyde (Sigma-Aldrich). Samples were run on a

Becton Dickinson Canto II with FACSDiva software (version 8.0). Data was analyzed with FlowJo software (version 9.8.8, BD Biosciences). Antibodies used include: CD4-FITC (1:30, SK3, BD 340133), CD8-PerCP-Cy5.5 (1:100, SK1, BD 341051), CD14-APC (1:100, M5E2, BD 561708), CD15-FITC (1:100, HI98, BD 560997), and CD16-PE (1:50, 3G8, BD 555407). The epidermis in a skin-on-chip was infected with $10^6$ PFUs of K26 for 1 h, while 300,000 neutrophils were perfused through the microvascular network. The live cell images were captured by a high-resolution Photometrics Coolsnap HQ2 scientific CCD camera equipped on a Nikon Ti microscope. The skin-on-chip was fixed 5 h or 20 h later for assessment by immunofluorescent staining.

For IL-8 neutralizing experiments, 5 μg/ml of human IL-8 neutralizing antibody (R&D systems, MAB208) was perfused through the vessel and added into the dermis 6 h before HSV infection. Mock treatment was performed with media only. In total, 300,000 human fresh isolated neutrophils were perfused through the vessel, while $2 \times 10^5$ PFUs of K26 was added on top of the epidermis. One day after infection, the skin-on-chip was fixed by 4% paraformaldehyde, embedded into OCT, snap frozen, and cryo-sectioned into 10 μm slides, and assessed by immunofluorescent staining.

**Cytokine measurement in dermis conditioned medium**
Cytokine measurements were performed by the Immune Monitoring Lab at Fred Hutchinson Cancer Center using Luminex multiplex assay. Conditioned medium was collected through the dermis access port of the device and stored at −20 °C until use. Cytokine-specific Luminex microbeads, one unique bead population per cytokine, were incubated with experimental samples and cytokine standards. Beads were then washed and incubated with biotinylated cytokine-specific antibodies, followed by another wash, and then incubated with a fluorescence-labeled streptavidin conjugate. After a final wash, samples were read on a Luminex 200 instrument. A standard curve is generated for each cytokine and sample concentrations are calculated from these curves.

**Diagram software**
Figures 1a, b, 3a, and Supplementary Figs. 1, 2 were created with SolidWorks (version SP3).

**Statistics and reproducibility**
GraphPad Prism (version 9.0.2, GraphPad Software) was used for all statistical analyses. Significance of two-group comparisons was determined by two-tailed Student's $t$ test. Significance of multiple comparisons between more than two groups was analyzed by two-way ANOVA. Differences at $p < 0.05$ were considered statistically significant. Figures were prepared using GraphPad Prism. All data points were derived from two or more biological replicates, as indicated for each experiment. Each image in Figs. 1d–j, 2b–e, 3b–e, 4b, 5e, 6a, c–e, 7d, f, S3, S4a, b are representative of experiments that were repeated independently at least two times.

**Reporting summary**
Further information on research design is available in the Nature Research Reporting Summary linked to this article.

## Data availability
The data that support the findings of this manuscript can be found in the main article file, Supplementary files, and Source data file. Source data are provided with this paper.

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

## Acknowledgements

We acknowledge Dr. Lawrence Corey for valuable discussions, Dr. Zhenhua Alvason Li for supporting with 3-D data analysis, and Dr. Amanda Woodward Davis for editorial assistance. This research was supported by the Cellular Imaging Shared Resource (CISR) of the Fred Hutch/University of Washington Cancer Consortium (P30 CA015704). We acknowledge the Immune Monitoring and Flow Core facilities in the shared resources of Fred Hutchinson Cancer Research Center for cytokine measurement and flow cytometry support, respectively. Figures 4a, 7a were created with BioRender.com. This work was supported by National Institutes of Health grant AI143773 and TR003208 to J.Z.

## Author contributions

S.S. designed research procedures, conducted experiments, analyzed data, and drafted the manuscript. L.J. performed tissue staining and imaging acquisition. Y.Z. developed microvascular engineering technology. J.Z. conceived the study, supervised the experiments, analyzed and interpreted the data, led writing of the manuscript, and provided funding support to the study. All authors contributed to the discussion and revision of the manuscript.

## Competing interests

The authors declare no competing interests.
