## [Peer Review File · Nature Communications]

Reviewers' Comments:

Reviewer #1:

Remarks to the Author:

For the authors

General comment: Decline

The study describes a vascularized and perfused skin-on-chip model which has been used to investigate virus infection and targeting by a drug. The manuscript is very interesting and adds to the field of organ-on-chip and skin infection models, but needs to be played down a bit since a number of claims are overstated or unfounded. Numerous controls are missing. The manuscript describes a combination of their published manuscript by Zheng 2012 (ref 49, 79) describing the vascularized chip and the model described by many scientists on a reconstructed epidermis on a fibroblast populated collagen hydrogel. Skin on chip has also been described by others, (see review: Progress and Future Prospectives in Skin-on-Chip Development with Emphasis on the use of Different Cell Types and Technical Challenges. van den Broek LJet al. Stem Cell Rev Rep. 2017 Jun;13(3):418-429. doi: 10.1007/s12015-017-9737-1. Review. This review was published in 2017 and since then further advancements have been made. None of this work by others is acknowledged and therefore does remove some of the novelty of the study. The new part of the study is the perfusion and invasion of neutrophils into the hydrogel and the targeting by a drug. Most key data is obtained from representative images. To make hard conclusion other methods should also be included e.g. pfu's to quantify number of virus obtained from infected skin model at different time points.

Abstract

The abstract overstates the results and is extremely generalized. For example the skin on chip does not represent natural skin architecture as it only contains a bilayered construct consisting of epidermal keratinocytes, fibroblasts and endothelial cells. The skin is a much more complex organ. The mechanical stimulation is a non-standardized incision introduced by a dermatology punch in which depth is not controlled. Mechanical stimulation indicates shear stress or pressure. Neutrophil activation is not shown, but migration. The results do not show how robust the model is (line 33) as no intra or inter experimental replicates are shown. The data consists mainly of representative images and graphs are also representing data from these images.

Introduction

As with abstract, it needs to be down played a bit. State of the art needs to be included. P 4; Organ on chip field is just starting, it has lots of potential but also lots of limitations and hurdles which need to be overcome. The skin as a complex organ is described consisting of epidermis, dermis, adipose tissue and appendages etc. Then you describe a simple bi-layered model as a full thickness skin equivalent for the rest of the manuscript. This needs down playing.

Results

P5: a collagen 1 rat tail hydrogel is used to represent the dermis this is not a "proper extracellular matrix".

What exactly F-actin staining. This is not clear in the images and text.

The use of the word "validated" is overstated. The model is tested with a single drug and 2 virus. No further data is shown on controls, replicates, reproducibility etc.

It is unclear from the text whether nutrients only reach the skin model via perfusion of the microvessels or also via traditional air – liquid exposure. Also what the times of perfusion are. How long is the model stable: days, weeks. This should be illustrated with analysis of LDH, glucose uptake, lactate production in the perfused culture medium for at least 1-2 weeks.

For the characterization, proliferation marker Ki67, granular layer markers SPPR genes, loricrin, involucrin, filagrin should be included. Currently the limited histology, K14 and K10 expression shown it seems to indicate a senescing culture with aberrant intermittent K14 (basal) and K10 (suprabasal) expression instead of confluent expression basally and suprabasally respectively as in native skin.

P6: The microvessels were perfused with Dextran-FITC for 3 mins. This time is extremely short to make the statement that the endothelial cell layer is barrier competent. Maybe the same result

would be achieved in the absence of endothelial cells considering the high concentration of the hydrogel used. This experiment should be repeated for longer times and with the control, without endothelial cells to make this statement.

P7: it is unclear what experiments represent static and what experiments represent dynamic perfused flow during the HSV viral infection.

Furthermore gene expression was not studied but protein expression. Vimentin appears lower than in native skin and in skin blood vessels stain positive as well with vimentin. K14 seems high and K10 less and intermittent more representative of a moist mucosa phenotype than a dry air exposed phenotype from the image in fig 2.

P8: the method of applying the virus should be written more clearly. As far as I understand, a dermatology punch was used to make a circular wound of non standardized depth. The epidermis was not removed. The virus penetrated via the incision circle introduced by the sharp punch. Mechanical disruption could indicate tape stripping of the stratum corneum.

Is it already known and shown that HSV infect only proliferating keratinocytes – if so this is not a new finding.

P8: Neutrophils are perfused into the microfluidics. Please characterise and show the purity of the cells by eg FACS analysis and relevant biomarkers to indeed show that they are neutrophils. The statement that the endothelialized microvasculature demonstrates proper function can only be said if the FITC perfusion experiment requested shows barrier competence. It could be that neutrophils are migrating into the gel where endothelial cells are absent.

Line 183: what is meant by post perfusion, wasn't perfusion continuous?

Line 184: cytokines were not analysed real time. A single time point was measured from collected supernatant.

P10: the claim that keratinocytes make enough IL8 to result in neutrophil migration into the hydrogel is unfounded and not shown. Indeed it has been reported that keratinocytes make much less IL8 (and IL6) than fibroblasts in such skin models and that it is keratinocyte derived IL-1 α which triggers fibroblasts to produce chemokines in large quantities. Does the virus infection result in IL-1 α release from keratinocytes. This needs to be measured at mRNA level or in absence of fibroblasts as fibroblasts will internalize it quickly.

See Spiekstra SW, et al., *Exp Dermatol.* 2005 Feb;14(2):109-16.

Boxman I, et al *J Invest Dermatol.* 1993 Sep;101(3):316-24.

Was ACV cytotoxic to neutrophils (or any of the cells in the model), this could be shown with eg flow cytometry and PI / annexin 5 staining. This is an important missing control.

TNF α is not a chemokine for recruitment but a pro-inflammatory cytokine

P11: again I miss control cytotoxicity data. Does ACV kill cells, can be measured eg with LDH, lactate release.

Again control without endothelial cells is needed to make statements in text.

Discussion:

This needs rewriting when results have been revised

P13: the platform is not a low cost model when taking into account specialized lab and technician work hours.

Line 250: you claim that you identified that undifferentiated basal keratinocytes are targeted by HSV, but this is already known (see introduction)

IL-8 is not primarily secreted by keratinocytes and you do not show this either

You show representative data, this does not show a robust model, but a very promising model

I miss a section on limitations and future perspectives. How is it better than other static and dynamic skin models which have been published.

Materials and Methods

P16 and 17: what body location was the skin derived from?

P17: for how long were the microvessels perfused?

How long after pouring the hydrogel were the endothelial cells added?

P18: how did you visualize 100% confluency of the keratinocytes in such a system?

It is not clear when static and dynamic culture conditions were used?

Reviewer #2:

Remarks to the Author:

In the paper, the authors detail the development of an in vitro 'skin-on-chip' composed of stratified multi-layered keratinocytes overlying a dermal fibroblast/collagen layer with an embedded endothelial cell lined venule-like vascular network to mimic human skin architecture. The vascular network can be perfused with immune cells (e.g. neutrophils) and drugs (e.g. acyclovir). The authors utilize this skin-on-chip device to model a human herpes simplex virus (HSV) infection and demonstrate 1) typical features of cutaneous HSV infections, 2) neutrophil extravasation and directional migration during infection, 3) key cytokines that trigger neutrophil activation during infection, and 4) acyclovir inhibition of HSV infection in a dose-dependent and time-sensitive manner.

While the application of skin-on-chip devices for the study of HSV infection is an interesting tool to understanding cutaneous HSV infection, especially early host-immune interactions that are poorly understood, the development of vascularized skin-on-chip devices is not itself novel. Several groups have developed similar devices (Groeber et al, ALTEX, 2016; Abaci et al, Adv Healthcare Mater., 2016; Mori et al, Biomaterials, 2017; Lee et al, Biomed Microdevices, 2017). These papers and other prior literature are not discussed in the current paper or contrasted against the current device.

While this device has the potential to be an interesting model for others in the field, it is not clear that knowledge presented in the paper will influence thinking in the field. There are several human skin elements that are lacking from the current model, including resident memory T cells (which reside in great numbers in the skin, Clark et al, J Immunol 2006), antigen presenting cells (especially Langerhans cells), neurons, hair follicles, and adipocytes. While current in vitro technology may not allow for a model with all these elements, these limitations are not addressed by the authors nor is it discussed how the absence of these elements may influence the validity of the model.

For example, the authors highlight an interesting limitation of their model by contrasting the expression of RANTES and TNF- α in their device (low) to that in vivo (high). The authors state "other immune cell types, however, might be responsible for secreting RANTES and TNF- α " and "the skin-on-chip platform might provide a useful in vitro system to delineate complex in vivo human immune responses to HSV infection". It would be interesting and informative if the authors explored these more complex immune responses. Perhaps a first step could be to delineate which cell types produce these cytokines in an HSV infection and determine if it is feasible to incorporate them into the skin-on-chip model.

In general, the manuscript is well written, and the authors do not overstate their claims. However, the authors do not present the current device in the context prior literature and the limitations of their model are not discussed. Additional experiments exploring the complex host immune and viral response would make this paper more influential to the field.

Response to reviewers

We thank the reviewers for the encouragement and interest in our work. The comments were thoughtful and insightful, and they contributed greatly to improve the quality of our manuscript. In response, we have generated more experimental data and revised the manuscript. Our revision can be summarized in the following areas:

1. We have revised the Abstract with more accurate and detailed description of the design and results, making sure not to overstate our findings.
2. The Introduction has been rewritten to better place our work in the context of the current literature and state-of-the-art technologies in the field of skin constructs and vascularization.
3. Expanded characterization of the multilayered epidermis in 3-D skin-on-chip platform using additional markers (filaggrin, loricrin, involucrin and Ki67). The results are presented in the new Supplementary Figure 3.
4. Permeability studies to include acellular microchannels for proper experimental control in the Dextran-FITC perfusion assay. Please see revised Figure 1H, the new Supplementary Figure 5, and the new supplementary movies 1 and 2.
5. Evaluation of the cellular sources of IL-8, RANTES and TNF- α expression using combined fluorescence in situ hybridization and immunohistochemistry method in the biopsy tissue of human genital herpes lesion. Please see the new Figure 6 (note: the original Figure 6 is now Figure 7 in the revision).
6. Modified the Discussion to add description on the unique design and advanced features of our device. We also dedicated a paragraph on the limitation of our current model with future directions.

We sincerely appreciate all the comments brought up by the reviewers, individual comments/questions have been addressed below.

Reviewer #1

1. The reviewer thought our study “is very interesting and adds to the field of organ-on-chip and skin infection model” but wanted us to play down a bit of overstated claims and add missing controls.

We thank the Reviewer #1 for the expert review and helpful comments. We have revised the manuscript to describe the model more accurately and provide more detailed information. We have eliminated overstated wording in the Abstract (page 2, lines 25-34) and throughout the text. We have added control data to several experiments, including the acellular microvessel network as a negative control for permeability analysis (pages 7-8, lines 132-134, Supplementary Figure 5, Supplementary Movie 1 and 2); additional proof of biomarker expression on stratified skin (page 7, lines 115-119), Supplementary Figure 3); and *in situ* evaluation of *IL8*, *RANTES* and *TNFA* mRNA transcript expression in genital herpes lesion and normal control skin (page 12, line 230-238, Figure 6C-E).

2. The reviewer pointed out we have missed key reviews in the literature of skin on chip (such as “Progress and Future Prospective in Skin-on-Chip Development with Emphasis on the use of Different Cell Types and Technical Challenges” by Van den Broek LJ et al) and it affected the novelty of the current study.

We sincerely appreciate the Reviewer’s criticism and apologize for being so focused on our own model in the prior manuscript. In the revision, we have restructured the introduction to acknowledge the up-to-date literature on the advancement of skin-on-chip and vascularization. We have now cited important reviews and original works, adding a total of 19 new references in the manuscript, including the one suggested by the reviewer (page 3-5). Further, we have summarized the novelty and advances of our device in the discussion (page 14-19). These improvements have better positioned our work in the context of state-of-the-art knowledge in the skin-on-chip field.

3. Most key data is obtained from representative images. To make hard conclusion other methods should also be included.

A large portion of our data were acquired via live cell imaging or *in situ* visualization by confocal microscopy. When designing our skin-on-chip device, we purposefully planned it to have a total depth of 500µm. This microscopy-compatible distance, as well as the incorporation of a fluorescence-tagged virus, allowed us to capture live events of dextran perfusion and neutrophil migration, and to examine viral infection from fixed cells. We used maximal projected confocal images, which accounted for the entire skin-on-chip, for quantification of epidermal infection, neutrophil migration, and antiviral drug effects. In this way, we have captured, analyzed, and presented data of the entire area, not just representative images. Being compatible with confocal microscopy is one of the unique advances of our device.

We have also used quantitative PCR assays to measure the viral loads through genome copies to compare viral replication between different treatments/conditions. The PCR results were very

similar to the ones measuring viral GFP expression shown in the manuscript (page 13, lines 247-248). We now have added the PCR data as Supplemental Figure 8.

Detailed points:

4. Abstract

The abstract overstates the results and is extremely generalized. For example the skin on chip does not represent natural skin architecture as it only contains a bilayered construct consisting of epidermal keratinocytes, fibroblasts and endothelial cells. The skin is a much more complex organ. The mechanical stimulation is a non-standardized incision introduced by a dermatology punch in which depth is not controlled. Mechanical stimulation indicates shear stress or pressure. Neutrophil activation is not shown, but migration. The results do not show how robust the model is (line 33) as no intra or inter experimental replicates are shown. The data consists mainly of representative images and graphs are also representing data from these images.

We appreciate the reviewer's comments. We have made several changes to describe specifically and accurately the features of our current skin model in the abstract and discussion. For mechanical stimulation, we originally did refer to it as "flow shear stress" and its influence on the actin alignment of endothelial cells in the vessel wall (Figure 1i). However, we have realized that it was not the major point and focus of our paper, so we have removed it and used the wording of mechanical stimulation. We also removed activation for neutrophil and accepted the word "promising" to replace "robust" to describe our platform. Please see the revised abstract and its specific changes in lines 25-34.

5. Introduction

As with abstract, it needs to be down-played a bit. State of the art needs to be included. P 4; Organ on chip field is just starting, it has lots of potential but also lots of limitations and hurdles which need to be overcome. The skin as a complex organ is described consisting of epidermis, dermis, adipose tissue and appendages etc. Then you describe a simple bi-layered model as a full thickness skin equivalent for the rest of the manuscript. This needs down-playing.

We agree with the reviewer that it would be important to place our model in better context within the organ-on-chip field. The introduction (pages 3-5) and discussion (pages 14-19) have been extensively modified and we have been careful to reference manuscripts that have made significant contributions to the field. We understand the concerns of the reviewer in the use of the phrase 'full thickness skin equivalent', however, it is widely used in the literature to refer to the differentiated multilayered stratified epidermis and underlying dermis. Based on the literature we feel that our model fits well with other models that use this term. Importantly, our model includes functional endothelial microvessels embedded within the collagen-containing dermal layer. We hope to contribute to the skin-on-chip field using our expertise in fluorescence microscopy and many years' experience studying HSV and the immune response *in vivo* in human genital skin. Further, we agree that there are limitations to this model and expand our discussion (pages 17-18, lines 353-370) to include the limitations and future directions.

Results

6. A collagen 1 rat tail hydrogel is used to represent the dermis this is not a “proper extracellular matrix”.

We thank the reviewer for this question and would like to clarify it. The traditional skin equivalent constructs contain collagen at < 3mg/ml, but they suffer from substantial contraction and shrinkage induced by embedded fibroblasts. To maintain the stability of microvessel geometry and overcome the contraction issue, we engineered our skin-on-chip system with a stiffer collagen at 7.5mg/ml to allow high reproducibility of vessel microstructure. Rat tail has been reported to be a good source of extracellular matrix for skin-on-chip models (Song et al. *Fabrication of a pumpless, microfluidic skin chip from different collagen sources*. 2017). It also enables remodeling through degradation and deposition of extracellular matrix by embedded fibroblast cells. Importantly, it is not considered to be immunogenic. Therefore, we designed our platform with these constraints in mind, which proved to be a useful tool for modeling host immune responses to viral infection.

We have amended the text to read: “composed of a stratified epidermis, an underlying dermis with a collagen-rich extracellular matrix containing fibroblasts and a microvascular network.” (page 4, lines 91-93).

7. What exactly F-actin staining. This is not clear in the images and text.

We used Alexa Fluor conjugated phalloidin to label filamentous actin (F-actin) in fixed tissue to broadly visualize cells and, in some cases, their cytoskeleton alignment. Phalloidin is a type of toxin that can directly bind and stabilize F-actin. We have added this description in the legend of Figure 1 and to the text in line 123.

8. The use of the word “validated” is overstated. The model is tested with a single drug and 2 virus. No further data is shown on controls, replicates, reproducibility etc.

We appreciate the reviewer for their feedback and have removed this language from the results sub-heading. However, we would like to point out that the manuscript now includes additional skin differentiation markers (filaggrin, loricrin, involucrin and Ki67) to characterize the multilayered epidermis in our skin-on-chip platform and in native human skin (new Supplementary Figure 3). Although our original manuscript did contain some quantitative analysis, we have added more replicates to the current version. Please reference HSV infectivity in Figures 2&7, neutrophil migration in Figure 4, and cytokine expression in Figures 5&6.

9. It is unclear from the text whether nutrients only reach the skin model via perfusion of the microvessels or also via traditional air-liquid exposure.

We thank the reviewer for this question and agree it is important for this to be clearly explained. There are two reservoirs in direct contact with the dermis, two reservoirs (inlet and outlet) that connect to the microvessels (Supplementary Figure 2) and a middle reservoir for growing

epidermis. Nutrients reach the 3-D skin via both perfusion of the microvessels and directly from the middle keratinocytes culture for air-liquid exposure.

10. What the times of perfusion are. How long is the model stable: days, weeks, etc. This should be illustrated with analysis of LDH, glucose uptake, lactate production in the perfused culture medium for at least 1-2 weeks.

We apologize that in the original manuscript perfusion times were not written more clearly. The perfusion was achieved by gravity-driven flow which was refreshed twice a day. The microvessels were perfused for about 2 weeks before the HSV experiments were performed (page 21, line 424) and this was the longest period we maintained the skin-on-chip culture. Cultures appeared stable and viable based on the tissue morphology and successful replication of virus upon infection. We agree that it would be useful to have more information on the status of the culture system. Microsensors might be incorporated to help monitor various parameters suggested in the future design of skin-on-chip devices. However, we feel it is beyond the scope of the current study.

11. For the characterization, proliferation marker Ki67, granular layer markers SPPR genes, loricrin, involucrin, filaggrin should be included. Currently the limited histology, K14 and K10 expression shown it seems to indicate a senescing culture with aberrant intermittent K14 (basal) and K10 (suprabasal) expression instead of confluent expression basally and suprabasally respectively as in native skin.

We agree with the reviewer and have included experiments to assess expression of Ki67, Loricrin, involucrin and filaggrin in both our 3D skin and human native skin. These results have been added as the new Supplemental Figure 3 (page 7, lines 115-120).

12. The microvessels were perfused with Dextran-FITC for 3 mins. This time is extremely short to make the statement that the endothelial cell layer is barrier competent. Maybe the same result would be achieved in the absence of endothelial cells considering the high concentration of the hydrogel used. This experiment should be repeated for longer times and with the control, without endothelial cells to make this statement.

We apologize that this was not clearly written. We agree with the reviewer that an acellular control would be very useful and have included these results. We have perfused the microvessel network for 20 mins with and without endothelial cells in the channel. The results are shown in the new Supplemental Figure 5 and in the new Supplemental Movies 1 & 2. We measured the vessel permeability between minute 7 and 10, so the measurement period is 3 mins, but the entire perfusion time is 20 mins. The experiments are summarized in the manuscript (pages 7-8, lines 131-136). We have also clarified the protocol in the Methods (page 22, lines 447-458).

13. It is unclear what experiments represent static and what experiments represent dynamic

perfused flow during the HSV viral infection. Furthermore gene expression was not studied but protein expression. Vimentin appears lower than in native skin and in skin blood vessels stain positive as well with vimentin.

We thank the reviewer for these comments. All our 3-D skin models were perfused by gravity-driven flow which was refreshed every 12 hours. Overall, we find very similar expression patterns of these key markers between our ‘skin-on-chip’ and native human genital skin. Fibroblast cells were shown to cap the vessel, which could result in positive staining (Supplementary Figure 1g). Of note, vimentin was previously reported to express on endothelial cells (Dave JM & Bayless KJ. *Vimentin as an integral regulator of cell adhesion and endothelial sprouting*. 2014).

14. The method of applying the virus should be written more clearly. As far as I understand, a dermatology punch was used to make a circular wound of non-standardized depth. The epidermis was not removed. And The virus penetrated via the incision circle introduced by the sharp punch. Mechanical disruption would indicate tape stripping of the stratum corneum. Is it already known and shown that HSV infect only proliferating keratinocytes – if so this is not a new finding.

We thank the reviewer for this question and have amended the Results (page 8, lines 158-166) and Methods (page 25, line 522-524) for clarity. The reviewer is right that the epidermis was not removed, virus entered through the incision circle. While it is known that HSV replicates in proliferating keratinocytes, it has not been shown that differentiated epidermal layer is not permissive for HSV infection. This specific experiment provided proof that HSV infection must initiate in proliferating keratinocytes.

15. Neutrophils are perfused into the microfluidics. Please characterize and show the purity of the cells by FACs analysis and relevant biomarkers to indeed show that they are neutrophils.

We thank the reviewer for this question, and we agree it is very important to demonstrate the purity of this population, which we have done in Supplemental Figure 7. We analyzed the isolated neutrophils by flow cytometry and found high expression of CD15 and CD16, but lacked expression of CD4, CD8 and CD14.

16. The statement that the endothelialized microvasculature demonstrates proper function can only be said if the FITC perfusion experiment requested shows barrier competence. It could be that neutrophils are migrating into the gel where endothelial cells are absent.

We thank the reviewer for this question. We have included an acellular control in the FITC experiment and compared the dramatic difference between acellular versus endothelialized microvasculature. We were able to clearly show the barrier competence in Supplementary figure 5 and in Supplementary Movies. Also, neutrophil transmigratory activity was completely blocked when perfused with an IL-8 neutralizing antibody. This evidence all supports the notion

that neutrophils were actively migrating and was not due to the absence of endothelial cells in the vasculature.

17. Line 183: what is meant by post perfusion, wasn't perfusion continuous?

The perfusion is continuous. We have amended the text to read "20 hours after the initiation of perfusion" (page 10, line 199).

18. Line 184: cytokines were not analyzed real time. A single time in point was measured from collected supernatant.

We thank the reviewer for this comment and have removed the phrase "real-time" from the text.

19. The claim that keratinocytes make enough IL8 to result in neutrophil migration into the hydrogel is unfounded and not shown. Indeed it has been reported that keratinocytes make much less IL8 (and IL6) than fibroblasts in such skin models and that it is keratinocyte derived IL-1a which triggers fibroblasts to produce chemokines in large quantities. Does the virus infection result in IL-1a release from keratinocytes. This needs to be measured at mRNA level or in absence of fibroblasts as fibroblasts will internalize it quickly.

We thank the reviewer for this question and have performed additional experiments to assess IL-8, RANTES and TNF protein (Figure 5 and pages 10-11, lines 198-208) and RNA expression (Figure 6b and page 11, lines 219-222). First, we assessed a natural herpes lesion for IL-8 expression and neutrophil migration and found that infected keratinocytes and those immediately surrounding the lesion expressed *IL8* RNA transcripts using a combined FISH and IHC method. Abundant *IL8* transcripts were detected in K10+K14- keratinocytes in addition to K14+ cells (Figure 6c and page 11-12, lines 222-226), suggesting differentiated keratinocytes might be a major source of IL-8 production upon viral infection. We also observed IL-1a transcripts release from keratinocytes (data not shown). We emphasized the importance of intact 3-D skin structures in evaluating viral infection and cytokine expression (page 17, lines 333-338). This might be the reason underlying the observed differences in cytokine expression between 2D and 3D cultures.

20. Was ACV cytotoxic to neutrophils (or any of the cells in the model), this could be shown with eg flow cytometry and PI / annexin 5 staining. This is an important missing control. TNFa is not a chemokine for recruitment but a pro-inflammatory cytokine

We appreciate this question but would like to clarify that we did not apply ACV in any neutrophil experiments.

21. Does ACV kill cells, can be measured e.g. with LDH, lactate release.

ACV and its prodrugs are widely used and considered to be safe and routinely applied in treating all herpes virus infections, even in immune compromised hosts, like transplant patients. It is well known that ACV has minimal toxicity to uninfected cells, because its antiviral activity relies on the expression of the viral thymidine kinase gene (King, DH. *History, pharmacokinetics, and pharmacology of acyclovir*. 1988).

22. Discussion needs rewriting when results have been revised

We appreciate this feedback and have made substantial changes to the discussion, which we believe has greatly improved the manuscript (pages 14-18).

23. The platform is not a low-cost model when taking into account specialized lab and technician work hours.

We thank the reviewer for making this point and have changed the wording to say “cost-efficient” (page 14, line 275). Although this model does require specialized lab equipment and trained technicians, like all experimental models, it has the potential to reduce overall costs compared to human clinical trials and animal models. In general, organ-on-chip models are significantly less expensive than human or animal studies and can be manipulated to gain valuable information.

24. Line 250: you claim that you identified that undifferentiated basal keratinocytes are targeted by HSV, but this is already known (see introduction)

In the introduction, we mentioned our prior work characterizing the spatiotemporal distribution of CD8+T cells (Ref. 10-14). There we found persistent CD8+ T cells localized at the dermal-epidermal junction in direct contact with basal keratinocytes where sensory nerve termini are found. The anatomic localization implied basal keratinocytes are the targets of reactivated viruses, which has been a point of interest for us. In Figure 2, we provided experimental evidence to show that basal keratinocytes were most susceptible to HSV compared to other layers.

25. IL-8 is not primarily secreted by keratinocytes and you do not show this either You show representative data, this does not show a robust model, but a very promising model

Please see the responses to question #19. We have shown in Figure 6C that both basal keratinocytes and differentiated keratinocytes express *IL8* mRNA transcripts in a human genital skin herpes lesion. Although IL-8 can be expressed by fibroblast cells in the dermis, data shown in Figure 6C suggest the majority of *IL8* transcripts are from the infected epidermal keratinocytes. Interestingly, the highest level of *IL8* expression was detected in K10+K14-keratinocytes surrounding the lesion. The results are described and discussed (page 11-12 and 16-17, respectively)

We have used the word “promising” to replace “robust” in the abstract (line 33).

26. I miss a section on limitations and future perspectives. How is it better than other static and dynamic skin models which have been published.

We thank the reviewer for bringing this to our attention and have added two sections to the discussion, one describing the advances of our model (pages 14-15), while the other covered the limitations of this model and future perspectives (pages 17-18).

Materials and Methods

27. What body location was the skin derived from?

The biopsies were derived from genital skin in study participants with culture proven genital HSV-2 recurrence, we have updated the Methods section to include this information (page 22, line 443)

28. For how long were the microvessels perfused?

The microvessels were perfused for at least 2 weeks before the HSV experiments were performed (page 21, line 424).

29. How long after pouring the hydrogel were the endothelial cells added?

The endothelial cells were added the following day after the hydrogel was poured to allow the embedded fibroblasts to settle. We have updated the Methods to include this information (page 20, line 417).

30. How did you visualize 100% confluency of the keratinocytes in such a system?

To maintain sterility of the skin-on-chip, confluency was not regularly visualized. However, confluency had been carefully determined through experimentation so that we could seed the skin-on-chip and know that it would be confluent 2 days later (page 21, line 429).

31. It is not clear when static and dynamic culture conditions were used?

For all experiments, perfusion was achieved by gravity-driven flow which was refreshed twice a day (page 21, line 423).

Reviewer #2

1. In the paper, the authors detail the development of an in vitro ‘skin-on-chip’ composed of stratified multi-layered keratinocytes overlying a dermal fibroblast/collagen layer with an embedded endothelial cell lined venule-like vascular network to mimic human skin architecture. The vascular network can be perfused with immune cells (e.g. neutrophils) and drugs (e.g. acyclovir). The authors utilize this skin-on-chip device to model a human herpes simplex virus (HSV) infection and demonstrate 1) typical features of cutaneous HSV infections, 2) neutrophil extravasation and directional migration during infection, 3) key cytokines that trigger neutrophil activation during infection, and 4) acyclovir inhibition of HSV infection in a dose-dependent and time-sensitive manner.

While the application of skin-on-chip devices for the study of HSV infection is an interesting tool to understanding cutaneous HSV infection, especially early host-immune interactions that are poorly understood, the development of vascularized skin-on-chip devices is not itself novel. Several groups have developed similar devices (Groeber et al, ALTEX, 2016; Abaci et al, Adv Healthcare Mater., 2016; Mori et al, Biomaterials, 2017; Lee et al, Biomed Microdevices, 2017). These papers and other prior literature are not discussed in the current paper or contrasted against the current device.

We sincerely appreciate Reviewer #2 for the precise summary, expert review, and helpful feedback. It is a critical point that we should put our model in the context of current skin-on-chip literature. In the revised manuscript, we have added a total of 19 new references related to state-of-the-art reviews and important original works in skin-on-chip and vascularization, including those suggested by Reviewer #2. To better highlight the unique designs and novel features of our model, we have designated a section in the discussion to detail the advances (pages 14-15, line 281-303 of our skin-on-chip platform).

2. While this device has the potential to be an interesting model for others in the field, it is not clear that knowledge presented in the paper will influence thinking in the field. There are several human skin elements that are lacking from the current model, including resident memory T cells (which reside in great numbers in the skin, Clark et al, J Immunol 2006), antigen presenting cells (especially Langerhans cells), neurons, hair follicles, and adipocytes. While current in vitro technology may not allow for a model with all these elements, these limitations are not addressed by the authors nor is it discussed how the absence of these elements may influence the validity of the model. For example, the authors highlight an interesting limitation of their model by contrasting the expression of RANTES and TNF- α in their device (low) to that in vivo (high). The authors state “other immune cell types, however, might be responsible for secreting RANTES and TNF- α ” and “the skin-on-chip platform might provide a useful in vitro system to delineate complex in vivo human immune responses to HSV infection”. It would be interesting and informative if the authors explored these more complex immune responses. Perhaps a first step could be to delineate which cell types produce these cytokines in an HSV infection and determine if it is feasible to incorporate them into the skin-on-chip model.

These are very important points. We agree that limitations should be discussed further in the manuscript and have added a section to the discussion (pages 18, lines 354-370). It is necessary to consider the purpose of each model and how they can best be used to ask specific questions. As Reviewer #2 indicated, our current model could not include all the important cell types, such as adipocytes, neurons, hair follicles and Langerhans cells to address functions related to sensation, thermoregulation, and antigen presentation. Nevertheless, our contribution to the skin-on-chip field is the incorporation of a functional, interactive, blood vessel to mimic a vasculature capable of immune-cell and drug perfusion for studying the dynamic host-pathogen interactions and antiviral efficacies.

The vascularized skin-on-chip allowed us to address early host immune responses in the context of primary HSV infection. Specifically, we deciphered the relationship between keratinocytes and neutrophils, and the role IL-8 played in mediating neutrophil migratory responses, which can only be studied in the human system because rodents lack the *IL8* gene. To explore the sources of IL-8, *RANTES* and *TNF α* secretion, we conducted *in situ* analysis combining FISH and IHC methods in biopsy tissue bearing a genital herpes lesion. The newly obtained data is shown in Figure 6C-E (pages 11-12, lines 222-238). Our data suggested that keratinocytes are the major sources of IL-8 expression. However, *RANTES* can be detected abundantly in some inflammatory immune cells but at low levels in basal keratinocytes. This result supports the expression pattern of *RANTES* seen in our skin-on-chip model, where it was increased over control, but was expressed at much lower levels than *IL8* and *IL6*. It also suggests that other innate infiltrating cells might be the major source of this cytokine. *TNF α* expression varied widely among study participants as indicated in Figure 6B. The *in situ* staining data in Figure 6E suggested that $CD3^+$ T cells and other cell types in the epidermis expressed *TNF α* . Future researches aiming to improve the system and create more complex model are needed to delineate the cascade of early immune activation. While our current model is suitable for studying primary infection of HSV, subsequent development of an autologous system with HLA matching structure and blood components might lead to the establishment of tissue-resident memory immunity in the skin-on-chip platform. The autologous skin-on-chip platform might be critical in assessing the role of tissue resident memory T cells in human skin health and diseases, although it is beyond the scope of the current work.

3. In general, the manuscript is well written, and the authors do not overstate their claims. However, the authors do not present the current device in the context prior literature and the limitations of their model are not discussed. Additional experiments exploring the complex host immune and viral response would make this paper more influential to the field.

We appreciate the reviewer's positive feedback and insightful comments. The introduction (pages 3-5) and discussion (pages 14-18) have been extensively modified to carefully cite publications that made important contributions in the field and to add in discussion the limitations of our current device. In our previous tissue-based studies (Ref. 10-14), we have characterized resident memory $CD4^+$ and $CD8^+$ T-cell immune responses and their influences on the surrounding microenvironment in human genital skin during HSV recurrences. We intend to interrogate more closely with our skin-on-chip platform in the near future. While described in

this manuscript is our first vascularized skin-on-chip model, we foresee future generations using autologous cellular components to study immune responses in a personalized skin-on-chip setting. Successful isolation of primary cells: keratinocytes, fibroblasts, and endothelial cells simultaneously from the same skin biopsy or utilization of stem cell technology, along with matched PBMC, will allow investigations to address more complex questions related to the protective T-cell immunity and the underlying mechanisms of a wide spectrum of HSV disease outcomes.

We hope all the revisions we have made to the manuscript have sufficiently put our model into better context while highlighting the flexibility and potential for complexity it offers for future investigations.

Reviewers' Comments:

Reviewer #1:

Remarks to the Author:

The manuscript has been very extensively revised and I consider it now acceptable

Reviewer #3:

Remarks to the Author:

This manuscript by Sun et al describes a novel 3-D "skin-on-chip" platform and provides proof-of-concept by exploring the initial events of HSV infection. While this system has the some of the same limitations as other 3-D culture systems (i.e. minimal resident immunosurveillance, no innervation, etc), the model presented here could help improve our understanding of keratinocyte biology in relation to viral infection and could serve as an alternative pre-clinical model for testing antiviral therapeutics for HSV. Detailed comments are below.

1. In Figure 2b, the epidermis at day (+1) after air exposure looks very similar to the images shown at day (-1), and yet there is a significant reduction in the number of virally infected cells. Is there an explanation for this? Is there a change in expression of viral receptors? Increased expression of antiviral mediators such as interferons? A more in-depth profile of the keratinocytes, especially as it relates to HSV infection, would be helpful.

2. While use of a biopsy punch ensures that virus will pass through the epidermal barrier, would a less extreme method of introducing virus also work for introducing HSV into the system (i.e. surface abrasion, etc)?

3. Please describe the "mock infection" in Figures 4 and 5 - is it a punch biopsy with PBS inoculation? This is not clear from methods, text or figure legend.

4. There is a discrepancy between the methods and legend for Figure 3 in the description of the infection protocol (one says 1.5mm punch with 10^6 PFU, the other says 2mm punch with 10^5 PFU).

Reviewer #4:

Remarks to the Author:

Sun et al present a skin-on-chip model with a HSV infection proof-of-concept.

Their microfluidic model provides technical advances over previous culture systems by incorporating independent ports for delivery to different compartments, stiffer Matrigel to reduce shrinkage, and avoids use of PDMS, which can absorb small molecules.

The paper is well-written and the figures attractive, although the data presented is overwhelmingly descriptive and representative. The lack of quantitation is disappointing and sometimes confusing, as is the presentation of only two data points for some experiments given that these models are supposedly convenient to use.

Specific issues noted:

- In the neutrophil experiments, there appears to be a lack of consistency in the number of cells migrating in response to HSV infection between Figure 4c and Figure 5f (~50,000 versus 1,500). Further, the number of cells presented in Fig 5c at different z-planes does not seem to add up the number of migrated cells described in 5f.

- It is unclear what "mock" refers to in neutrophil experiments. Was the tissue punched but no virus added? What happens when the vessel is perfused in a true negative control without tissue disturbance? What were the "mock" controls used for anti-IL8 and ACV delivery?

- It would be stronger to show multiple lines of evidence that neutrophil migration is dependent on IL-8 production by keratinocytes, such as use of an IL-8 knockout keratinocyte cell line.

- It is unclear in the HSV-GFP experiments whether the presence GFP-fusion protein is a useful

readout of active viral replication, since GFP produced prior to treatment may remain stable for some time even while new replication is effectively halted.

With regards to revisions, it appears the authors have added/changed a lot of text in response to reviewer's comments, providing a more comprehensive overview of the state of the art and how they have improved upon it. From a biological viewpoint, the insight provided by their infection experiments is minor and in many cases improperly controlled. It is my view that this manuscript may be a better fit for a more specialized journal.

Response to Reviewers (Second Round)

We appreciate all the expert review of this manuscript, especially the comments brought up by the new Reviewers #3 and #4. We have addressed all concerns that were raised. Please see our detailed responses below.

Reviewer #1

The manuscript has been very extensively revised, and I consider it now acceptable.

We thank Reviewer #1 for the endorsement of the manuscript.

Reviewer #3

This manuscript by Sun et al describes a novel 3-D "skin-on-chip" platform and provides proof-of-concept by exploring the initial events of HSV infection. While this system has the some of the same limitations as other 3-D culture systems (i.e. minimal resident immunosurveillance, no innervation, etc), the model presented here could help improve our understanding of keratinocyte biology in relation to viral infection and could serve as an alternative pre-clinical model for testing antiviral therapeutics for HSV. Detailed comments are below.

We thank Reviewer #2 for appreciating the potential of this vascularized skin-on-chip model in biomedical applications for basic skin research and preclinical drug evaluation. We agree that the ideal *in vitro* skin model for HSV research would include both immune system components and sensory nerve innervation, integrated into the epidermal and dermal compartment. Development of such complex, dynamic, and integrated skin-on-chip platforms are a significant challenge in the field. The first step towards a more completed model requires incorporation of a functional vascular network in order to successfully transport and direct immune cells and to serve as a scaffold for guiding the nerve network. The work described here represents a critical advancement for achieving the goal of building a biomimetic skin-on-chip model suitable for HSV research.

1. In Figure 2b, the epidermis at day (+1) after air exposure looks very similar to the images shown at day (-1), and yet there is a significant reduction in the number of virally infected cells. Is there an explanation for this? Is there a change in expression of viral receptors? Increased expression of antiviral mediators such as interferons? A more in-depth profile of the keratinocytes, especially as it relates to HSV infection, would be helpful.

We thank the reviewer for these questions and would like the chance to further discuss. We have included a figure here with an enlarged view of the original Fig. 2B & C (**Fig.1a**). Although the morphology at one day before or after air lifting looked very similar at first glance, there were changes in the formation of epidermis. Notably, the rough surface of the epithelium from day -1 became smooth and flattened at day +1. A second keratinocyte layer can also be seen at one day following air lifting, forming a likely suprabasal layer. H&E and fluorescent images both suggest that keratinocytes are initiating upward growth and differentiation even one day after air lifting. We have added the information into the results on page 9, line 162-165.

To understand how keratinocyte differentiation influences HSV infectivity and replication, we investigated expression patterns of HSV entry receptors, nectin-1 and HVEM, in healed human skin (**Fig. 1b**). Nectin-1 is known for mediating HSV infection in human cells of epithelium and neuronal origin, while HVEM serves as the principal entry receptor for human lymphoid cells but no other cell types^{1,2}. Nectin-1 appears to be uniformly and abundantly produced and evenly distributed throughout the

entire epidermis. No difference in nectin-1 expression was observed across basal and upper epidermal layers. On the other hand, HVEM expression was absent in all keratinocytes and was present only near blood vessels deep in the dermis, consistent with the literature for its expression on immune cells. Thus, the reduction in HSV infectivity at day +1 following air lifting was likely not due to lack of virus entry receptor expression in the upper epidermal layers.

Figure 1. a. Images showing skin-on-chip one day before (Day -1) and one day after (Day 1) air lifting. H&E (top) and DAPI (bottom) staining. **b.** Nectin-1 (magenta) and HVEM (green) expression in human skin biopsy. Right images are enlarged views of dashed box with or with DAPI (blue).

In our recently published study, we investigated various types of interferon (IFN) expression in normal human skin and in HSV-2 affected ulcer lesion and post-healed skin³. Type I IFNs were mostly undetectable in all types of normal and HSV affected skin. Nevertheless, T cell derived IFN- γ was predominantly produced in skin during HSV-2 reactivation. IFN- γ inducible factor 16 (IFI16), known for its role in sensing and suppressing HSV infection, was widely expressed in protected epithelium across all skin layers, but was rarely detected in normal control skin or actively HSV-infected keratinocytes. We observed similar IFI16 expression patterns across basal and upward strata. Therefore, an intracellular antiviral state was induced in protected keratinocytes, but there was no evidence supporting the hypothesis of an enhanced IFN antiviral pathway upon keratinocytes differentiation for viral resistance intrinsically.

To understand epidermis development along keratinocyte differentiation trajectory, large-scale transcriptomic profiles of single keratinocytes from human skin have been reported, using single-cell RNA sequencing technology⁴. In that study, clustering analysis revealed expected cell states and classified specific markers well conserved in human skin from various anatomic location. Notably, mitotic cells were enriched in DNA synthesis and proliferation genes, and were found only in basal and suprabasal layer, suggesting suprabasal keratinocytes might retain some proliferative capacity. These transcriptomic data aligned well with our observations on productive HSV infection in the basal and suprabasal layer, shown at day -1 and day 1 time point. Reduced viral infectivity in suprabasal cells might be due to their lower proliferation capacity than basal keratinocytes. However, many genes expressed with spatially restricted differentiation patterns have unknown functions⁴. Detailed molecular mechanisms distinguishing HSV infectivity between basal and suprabasal keratinocytes remain to be defined in future studies.

We have summarized these discussions and added it on page 16-17, line 312-316.

2. While use of a biopsy punch ensures that virus will pass through the epidermal barrier, would a less extreme method of introducing virus also work for introducing HSV into the system (i.e. surface abrasion, etc)?

It will be interesting to compare the effectiveness and consistency of different methods in disrupting epithelial barrier and introducing virus. Surface abrasion via tape stripping is a good suggestion. Alternatively, we could explore a non-injury route by adding virus directly through the dermal compartment underneath the epidermis. We plan to test these in future experiments, but they are beyond the scope of the current manuscript.

3. Please describe the "mock infection" in Figures 4 and 5 - is it a punch biopsy with PBS inoculation? This is not clear from methods, text or figure legend.

We apologize for not explaining it clearly. All mock infections were done by disrupting the surface with a biopsy punch and followed by media only inoculation. We have amended the methods, on page 27, line 534 and page 29, line 568- 569 to include this information.

4. There is a discrepancy between the methods and legend for Figure 3 in the description of the infection protocol (one says 1.5mm punch with 10^6 PFU, the other says 2mm punch with 10^5 PFU).

We thank the reviewer for catching this and have clarified the figure legends on page 40, line 805-806; page 41, line 819; page 43, line 831-833; page 44, line 843-845. All experiments used a 1.5mm punch. Infections were done using 10^6 PFU except in the case of the IL-8 antibody experiments in Figure 5e&f, where 10^5 PFU of HSV was used.

Reviewer #4

Sun et al present a skin-on-chip model with a HSV infection proof-of-concept. Their microfluidic model provides technical advances over previous culture systems by incorporating independent ports for delivery to different compartments, stiffer Matrigel to reduce shrinkage, and avoids use of PDMS, which can absorb small molecules. The paper is well-written and the figures attractive, although the data presented is overwhelmingly descriptive and representative. The lack of quantitation is disappointing and sometimes confusing, as is the presentation of only two data points for some experiments given that these models are supposedly convenient to use.

1. In the neutrophil experiments, there appears to be a lack of consistency in the number of cells migrating in response to HSV infection between Figure 4c and Figure 5f (~50,000 versus 1,500). Further, the number of cells presented in Figure 5c at different z-planes does not seem to add up the number of migrated cells described in 5f.

We thank Reviewer #4 for careful reading and bringing this issue to our attention. As indicated above, all experiments used 10^6 PFU for HSV infection, except for the later anti-IL-8 neutralization experiments in Figures 5e&f, where 10^5 PFU was used for infection. The 10-fold fewer viral inoculation dosage resulted in much lower number of migrated neutrophils. This contributed to the difference seen in total cell number between Figure 5e&f and Figure 4c&5c. Nevertheless, our conclusion on neutrophil migration is consistent throughout Figure 4 and Figure 5. We apologize for any confusion it might cause and have amended the figure legends on page 41, line 819; page 43, line 831-833; page 44, line 843-845 to clarify.

2. It is unclear what “mock” refers to in neutrophil experiments. Was the tissue punched but no virus added? What happens when the vessel is perfused in a true negative control without tissue disturbance? What were the “mock” controls used for anti-IL8 and ACV delivery?

We primarily wanted to control for the effect of HSV infection, so all mock treatments were performed exactly the same as the infected group except using media in replacement of HSV. We did not perform control experiments where the vessel was perfused without epidermis disturbance as the primary focus was to control for the punch damage, since tissue damage can activate neutrophils and influence their behavior. For anti-IL-8 and ACV delivery, media alone was used as mock treatments. The comparisons were made between HSV+media and HSV+anti-IL-8 antibody, and again between HSV+media and HSV+ACV at different treatment time. We have amended the methods on page 27, line 534 and page 29, line 568- 569 to include the mock information.

3. It would be stronger to show multiple lines of evidence that neutrophil migration is dependent on IL-8 production by keratinocytes, such as use of an IL-8 knockout keratinocyte cell line.

We appreciate this suggestion. Generating various gene-specific knockout keratinocyte cell lines and integrating them into the skin-on-chip platform will allow us to study skin infection and human immune reaction in a mechanistic fashion. Currently, this is beyond the scope of the manuscript.

4. It is unclear in the HSV-GFP experiments whether the presence GFP-fusion protein is a useful readout of active viral replication, since GFP produced prior to treatment may remain stable for some time even while new replication is effectively halted.

The HSV-GFP recombinant virus, K26⁵ is a widely used tool in the HSV field for visualizing and monitoring viral infection *in vivo* and *in vitro*. GFP expression is under the control of the VP26 gene, a true late gene, whose expression is dependent on viral replication. Therefore, the detection of GFP signal correlates well with active viral infection and viral production. We usually detect GFP expression in primary cultured human keratinocyte 16 hours post infection (MOI=1) in a 24-hour experiment. Alternatively, PCR quantification offers a direct measure of viral loads, but it lacks the capacity to perform real-time monitoring for viral infection and live-cell screening for antiviral therapeutics. In Supplemental Figure 8 we used PCR-based detection for viral replication and found comparable results to Figure 7e using GFP as surrogate for viral replication.

References

1. Montgomery, R.I., Warner, M.S., Lum, B.J. & Spear, P.G. Herpes simplex virus-1 entry into cells mediated by a novel member of the TNF/NGF receptor family. *Cell* **87**, 427-436 (1996).
2. Geraghty, R.J., Krumpfenach, C., Cohen, G.H., Eisenberg, R.J. & Spear, P.G. Entry of alphaherpesviruses mediated by poliovirus receptor-related protein 1 and poliovirus receptor. *Science* **280**, 1618-1620 (1998).
3. Peng, T., *et al.* Tissue-Resident-Memory CD8(+) T Cells Bridge Innate Immune Responses in Neighboring Epithelial Cells to Control Human Genital Herpes. *Front Immunol* **12**, 735643 (2021).
4. Cheng, J.B., *et al.* Transcriptional Programming of Normal and Inflamed Human Epidermis at Single-Cell Resolution. *Cell reports* **25**, 871-883 (2018).
5. Desai, P. & Person, S. Incorporation of the green fluorescent protein into the herpes simplex virus type 1 capsid. *J Virol* **72**, 7563-7568 (1998).

Reviewers' Comments:

Reviewer #3:

Remarks to the Author:

Authors have addressed all comments satisfactorily.

Reviewer #4:

Remarks to the Author:

Thank you for addressing my concerns.

Response to Reviewers (Third Round)

We appreciate the reviewers support of our manuscript.

Reviewer #3

Authors have addressed all comments satisfactorily.

Reviewer #4

Thank you for addressing my concerns.